Deep learning-based novel ensemble method with best score transferred-adaptive neuro fuzzy inference system for energy consumption prediction

Dağkurs Birce 1
http://orcid.org/0000-0002-6357-0073 Atacak İsmail 2 iatacak@gazi.edu.tr
1 Graduate School of Natural and Applied Sciences, Advanced Technologies, Gazi University , Ankara , Turkey
2 IoTLab, Department of Computer Engineering, Faculty of Technology, Gazi University , Ankara , Turkey
Stević Željko
Electronic publication date: 2025 Feb 21
Publication date: 2025
Volume: 11
Electronic Location ID: e2680
Received 2023 Sep 27; Accepted 2025 Jan 13
Copyright: © 2025 Dağkurs and Atacak
Copyright year: 2025
Copyright holder: Dağkurs and Atacak
License: This is an open access article distributed under the terms of the Creative Commons Attribution License, which permits unrestricted use, distribution, reproduction and adaptation in any medium and for any purpose provided that it is properly attributed. For attribution, the original author(s), title, publication source (PeerJ Computer Science) and either DOI or URL of the article must be cited.
License URL: https://creativecommons.org/licenses/by/4.0/

Keywords: Deep learning, Machine learning, Adaptive neuro fuzzy inference system, Ensemble learning, Energy consumption

Funding: The authors received no funding for this work.

==============================
Background

Energy consumption predictions for smart homes and cities benefit many from homeowners to energy suppliers, allowing homeowners to understand and manage their future energy consumption, improve energy efficiency, and reduce energy costs. Predictions can help energy suppliers effectively distribute energy on demand. Therefore, from the past to the present, numerous methods have been conducted using collected data, employing both statistical and artificial intelligence (AI)-based approaches, to achieve successful energy consumption predictions.

Methods

This study proposes a deep learning-based novel ensemble (DLBNE) method with the best score transferred-adaptive neuro fuzzy inference system (BST-ANFIS) as a high-performance and robust approach for energy consumption prediction. The proposed method uses deep learning (DL)-based algorithms, including convolutional neural networks (CNN), recurrent neural networks (RNN), long short-term memory (LSTM), bidirectional long short-term memory (BI-LSTM), and gated recurrent units (GRUs) as base predictors. The BST-ANFIS architecture combines the individual outcomes of these predictors. In order to build a robust and dynamic prediction model, the interaction between the base predictors and the ANFIS architecture is achieved using a best score transfer approach. The performance of the proposed method in energy consumption prediction was verified through five DL methods, five machine learning (ML) methods, and a DL-based weighted average (DLBWA) ensemble method.

Results

In experimental studies, the results were obtained from three-stage analyses: fold, average, and periodic performance analyses. In fold analyses, the proposed method, in terms of the root mean square error (RMSE) metric, demonstrated better performance in four folds on the Internet of Things (IoT)-based smart home (IBSH) dataset, two in the homestead city electricity consumption (HCEC) dataset, and two in the individual household power consumption (IHPC) dataset compared to the other methods. In the average performance analyses, it showed significantly higher performance than the other methods in all metrics for the IBSH and IHPC datasets, and in metrics except the mean absolute error (MAE) metric for the HCEC dataset. The performance results in terms of RMSE, MAE, mean square error (MSE), and mean absolute percentage error (MAPE) metrics from these analyses were obtained as 0.001531, 0.001010, 0.0000031, and 0.001573 for the IBSH dataset; 0.025208, 0.005889, 0.001884, and 0.000137 for the HCEC dataset; and 0.013640, 0.006572, 0.000356, and 0.000943 for the IHPC dataset, respectively. The results of the 120-h periodic analyses also showed that the proposed method yielded a better prediction result than the other methods. Furthermore, a comparison of the proposed method with similar studies in the literature revealed that it demonstrated competitive performance in relation to the methods employed in those studies.

Introduction

Over the past few decades, global electrical energy consumption has grown rapidly due to population expansion and technological advancements (Kim & Cho, 2019b). According to the international energy agency (IEA) report (IEA, 2019), the world’s total electricity consumption increased from 12,137 terawatt-hours (TWh) to 22,848 TWh between 1999 and 2019. The report highlights that the residential and industrial sectors together contribute to 15,638 TWh, accounting for 66.45% of total energy consumption. The assessment emphasizes the rising energy consumption, especially in residential and industrial sectors, which requires urgent action from electricity distribution and supply companies (EDSCs) to develop effective policies and innovative solutions to manage current and future energy needs. In this context, experts have focused on developing tools that automate energy distribution, using load prediction methods to optimize energy source utilization (Acharya, Wi & Lee, 2019). These methods, known as innovative approaches, utilize various features such as historical data, weather patterns, and customer behaviors to predict future energy consumption. Accurate predictions enable EDSCs to manage energy supply effectively, enhance capacity, make capital investments, and conduct market research (Sajjad et al., 2020).

A comprehensive review of literature on energy consumption predictions reveals diverse studies employing different approaches. These studies are generally categorized into two classes: statistics-based methods and artificial intelligence (AI) based methods. Statistics-based energy consumption prediction methods are mathematical models that analyze historical data to predict future consumption trends (Box et al., 2016; Hyndman & Athanasopoulos, 2018; Cai et al., 2023). Methods such as Holt-Winters (HW) (Sudheer & Suseelatha, 2015), auto-regressive moving average (ARMA) (Chen, Wang & Huang, 1995), auto-regressive integrated moving average (ARIMA) (Büyükşahin & Ertekin, 2019), grey prediction (GP) (Li & Zhang, 2018), and seasonal auto-regressive integrated moving average (SARIMA) (Blázquez-García et al., 2020) tackle energy consumption prediction as a typical time series problem. These methods might yield successful results in linear predictions with regular data distributions, but their accuracy decreases in those with incomplete or unbalanced data distribution. Hence, AI-based methods, offering more robust and dynamic solutions, have gained prominence. Within these methods, especially machine learning (ML) and deep learning (DL) algorithms are widely used.

ML algorithms are essential for modeling complex relationships and predicting future energy consumption trends. Some popular ML algorithms for energy consumption prediction include classical methods, such as artificial neural network (ANN) (Kandananond, 2011), support vector regression (SVR) (Ma, Ye & Ma, 2019), and decision trees (DT) (Tso & Yau, 2007). Additionally, model-based ensemble learning (EL) techniques, such as random forest (RF) (Silva et al., 2020), Extreme Gradient Boosting (XGBoost) (Shin & Woo, 2022), and light gradient boosting machines (LightGBM) (Chowdhury et al., 2021), have shown promising results in energy consumption prediction. Classical ML-based methods can effectively model linear and nonlinear relationships but may have varying performance depending on the training data, especially with large datasets (Olu-Ajayi et al., 2022). On the other hand, model-based ensemble ML methods have demonstrated high performance, resistance to overfitting, and faster training times (except for the RF method), particularly with large datasets (Suenaga et al., 2023). However, these methods have limitations owing to the performance of the individual models and the variability of the input data. One limitation is the problem of high dimensionality, particularly in high-dimensional datasets with many attributes, which can lead to overfitting and poor performance (Tran, 2023). Another limitation is the less robust prediction problem caused by uncertainty and high variance in the input data in dynamic and variable environments (Shree et al., 2023). In addition, the aggregated outcomes may exhibit optimal performance on certain datasets, with variations in the accuracy levels of the model outputs depending on data characteristics.

DL-based methods are crucial for accurately predicting large and complex datasets. So far, recurrent neural network (RNN) and its variations, such as long short-term memory (LSTM), bidirectional long short-term memory (BI-LSTM), and gated recurrent unit (GRU), have been successfully used to predict energy consumption because of their ability to uncover hidden relationships over time (Fan et al., 2019; Bai et al., 2021). These methods outperform classical ML-based methods in predictions; however, their performance can vary depending on the data characteristics (Farsi et al., 2021). Furthermore, while DL-based methods may outperform ensemble ML methods in terms of prediction accuracy, they struggle to handle uncertainties in data from grid lines with sudden power changes and load imbalances. Consequently, hybrid and ensemble DL-based approaches have been proposed as effective solutions to these challenges.

Several studies have shown the effectiveness of hybrid DL-based methods, such as the convolutional neural network long short-term memory (CNN-LSTM) neural network (Kim & Cho, 2019b), CNN-multilayer bidirectional GRU (CNN-MB-GRU) neural network (Khan et al., 2020b), LSTM neural network with stationary wavelet transform (SWT) technique (Yan et al., 2019), CNN with LSTM autoencoder (LSTM-AE) (Khan et al., 2020a), and Novel CNN-GRU neural network (Sajjad et al., 2020), in accurately predicting electricity consumption. They have proven to be adept at capturing the temporal and spatial features that impact energy consumption and delivering accurate predictions. Each method has its strengths and weaknesses. The choice of the prediction method depends on the dataset’s characteristics, problem complexity, and learning objectives. However, these methods present common challenges, such as non-optimized parameter values, resource-intensive computations, and extensive data preprocessing, particularly in hybrid models incorporating the CNN method.

Although there have been relatively few studies on ensemble DL methods compared to hybrid DL-based methods in energy consumption prediction, their application has been steadily increasing in recent years. Studies in this area show that ensemble DL methods can be successfully applied to tackle energy consumption prediction tasks. The current ensemble DL methods proposed in the literature for this purpose include approaches that combine LSTM, GRU, and temporal convolutional networks (TCN) (Hadjout et al., 2022); LSTM and Kalman Filter (KF) methods (Khan et al., 2021a); LSTM and GRU with cluster analysis (Khan et al., 2021b); RNN, LSTM, BI-LSTM, GRU, and CNN (Shojaei & Mokhtar, 2024); LSTM, Evolutionary trees (EvTree), RF and neural networks (NN) (Sujan Reddy et al., 2022); ANN, XGBoost, LSTM, stacked LSTM, and BI-LSTM (Guo et al., 2024); and LSTM and RNN (Irfan et al., 2023). These methods leverage the powerful capabilities of DL models, such as their ability to learn hierarchical representations and capture complex patterns and relationships in energy consumption data. Depending on the data characteristics, they can reduce the risk of overfitting and improve generalization by combining multiple models. Ensemble DL methods are also more robust to noisy or biased data as they reduce reliance on a single model that may have been trained on potentially biased information. Despite the limited number of studies in the literature, average- or weighted-average-based ensemble DL methods are still used owing to their simple structures and ease of application. However, this simplicity has several disadvantages, such as low performance issues resulting from the loss of diversity among predictors, insensitivity to outliers, and the inability to capture complex effects based on the data structure or the characteristics of base predictors. To address these limitations, more sophisticated stacking ensemble DL methods have been proposed. These advanced ensemble techniques can leverage the complementary strengths of diverse DL models, leading to an improved predictive performance in energy consumption prediction. In stacking ensemble architectures, most studies show that ML methods, such as XGBoost, GBM, support vector machine (SVM), and k-nearest neighbor (KNN), can be successfully used to combine multiple DL predictor outputs on specific datasets. However, when these methods are used with high-dimensional datasets, they can encounter challenges, including long training times, overfitting, and generalization issues, which can result in an overall performance degradation. Furthermore, because the base predictors in these models are often complex, overfitting and generalization-related problems inherent in the individual models can further exacerbate these performance declines. To fill this gap in the literature, there is an urgent need to develop effective and robust models within the stacking ensemble DL framework that can minimize the problems associated with current prediction combiners. Designing such robust and high-performing meta-models is crucial for advancing the state-of-the-art research in this area.

In this study, the researchers address the gap in ensemble DL methods with an advanced stacking ensemble method specifically designed for household energy consumption data. In the proposed method, multiple DL models such as RNN, LSTM, BI-LSTM, GRU, and CNN are utilized as base predictors due to the short- and long-term dependencies and complex characteristics of such time series data. Unlike previous studies in the literature, the meta-combiner component of the method employs a configuration that effectively combines multiple DL outputs through Adaptive Neuro Fuzzy Inference System (ANFIS) using a best score transfer approach. The contributions of the proposed method to the literature are as follows. Combining the scores from the DL-based base predictors using an algorithm that incorporates human learning and inference abilities (ANFIS) has made the proposed method a more robust and dynamic approach compared to similar methods in the literature.

The innovative aspect of the proposed method, also known as its distinctive feature, is its ability to dynamically identify two models with the best score outputs among the base DL predictors and effectively combine these outputs through an optimized architecture. This characteristic of the method allows it to reduce the number of inputs to ANFIS, designing it as a simpler structure as a prediction combiner, while also enabling the effective combination of the best scores through this architecture. Structurally, this contributes significantly to reducing the complexity of the method and enhancing its performance as a prediction result. In addition, the best score effect can minimize performance issues arising from overfitting of base predictors in traditional stacking ensemble methods.

The model combines five robust DL-based methods (CNN, RNN, LSTM, BI-LSTM, and GRU) with the ANFIS architecture. CNNs are effective at feature extraction, while RNN and its variations (LSTM, BI-LSTM, and GRU) are skilled at capturing sequential dependencies in time-series data. This combination of methods enhances the model’s flexibility for energy consumption prediction.

The proposed method offers a new perspective to the literature by illustrating how DL models can be combined more effectively, resulting in substantial benefits.

The proposed method’s comprehensive analysis and comparison with existing DL, ML, and classical EL techniques clearly highlight its strengths and weaknesses, providing a valuable reference for future research.

This novel approach can also be applied to other time-series forecasting issues that have similar structures to energy consumption forecasting, such as water consumption and demand prediction, weather forecasting, and natural gas consumption prediction. Thus, the method’s scope can be expanded, making it suitable for use in various disciplines.

The remaining sections of this article are organized as follows. “Literature Review” provides a comprehensive literature review on energy consumption predictions. “Materials and Methods” presents the datasets used, the structure and components of the proposed method, the DL-based weighted average (DLBWA) ensemble method, DL and ML-based methods, and evaluation metrics. “Experimental Results and Discussion” discusses the experimental results of the proposed method, and the other methods employed for performance verification. The last section interprets the findings and offers recommendations for future studies.

Literature review

The prediction of energy consumption in smart homes and buildings has received significant attention in the literature because of its crucial role in energy resource planning and management. Various methods have been proposed and tested in real-time or on diverse datasets. This section provides an overview of adaptive, ML, DL, hybrid DL, and ensemble DL-based methods relevant to our study.

Adaptive methods aim to dynamically adjust and optimize energy consumption based on real-time data and feedback. Several studies indicate that ANFIS and its combined architectures have been successfully employed as adaptive methods for predicting building electrical energy consumption. The study conducted by Ghenai et al. (2022) utilized ANFIS to predict the energy consumption of educational buildings. The model was tested using publicly available electricity consumption data obtained from smart meters in office buildings in Washington, USA. The results from the experimental studies showed that R correlation coefficients of 0.97951, 0.9854, and 0.96778 were achieved at 0.5, 1, and 4-h periods, respectively. Adedeji et al. (2022) conducted a study comparing the performance of stand-alone ANFIS with a hybrid ANFIS optimized using the particle swarm optimization (PSO) method to predict energy consumption based on climatic factors in a multi-campus institution in South Africa. They validated the performance of both models using a self-collected dataset containing weather information. The experimental studies on four campuses demonstrated that the hybrid PSO-ANFIS model for campus D outperformed all other standalone and hybrid models for campuses A, B, and C, including a root mean square error (RMSE) of 0.147, mean absolute deviation (MAD) of 0.125, and MAPE of 2.89. In a similar study performed by Oladipo, Sun & Adeleke (2023), they proposed an ANFIS model optimized with the modified advanced particle swarm optimization (MPSO) technique to predict electrical energy consumption in a student dormitory at the University of Johannesburg. The study involves comparing this model with hybrid ANFIS models using different optimization algorithms. The models utilize meteorological variables such as wind speed, humidity, and temperature as inputs and electrical energy consumption as output. The proposed MPSO-ANFIS model outperformed other models, achieving the lowest values for RMSE (1.8928 KWh), MAD (1.5051 KWh), and a coefficient of variation (RCoV) of 0.1370 under the split conditions of 0.7. On the other hand, Alam & Ali (2023) presented for predicting energy usage in residential buildings during both normal and COVID-19 periods. Parameter optimization using the PSO method and data training via the subtractive clustering method were incorporated into the proposed method. They compared the performance of MANFIS-2 with stand-alone ANFIS, RF, and LSTM methods. MATLAB simulation results confirmed the effectiveness of MANFIS-2 in residential load prediction, with absolute average error (AVG) values of 1.2396, RMSE of 2.0398, and MAPE of 14.12%. MANFIS-2 surpassed stand-alone ANFIS, LSTM, and RF methods in energy consumption predictions for both pre-COVID-19 and post-COVID-19 conditions. Finally, Barak & Sadegh (2016) developed hybrid ARIMA-ANFIS ensemble models for annual energy consumption in Iran. The results from three hybrid patterns of ARIMA-ANFIS showed that the third hybrid pattern, using the adaptive boosting (AdaBoost) method with Genfis3 ANFIS structure, performed quite better with a MSE value of 0.026%.

Generally, the ability of ML algorithms to capture complex relationships, identify nonlinear patterns, generalize well, and adapt to various data types has made them considerably more effective in addressing a variety of regression problems than conventional methods. In this regard, several commonly utilized ML methods, as well as hybrid approaches, have been successfully employed to predict energy consumption in buildings. In a study conducted by Chammas, Makhoul & Demerjian (2019) a multilayer perceptron (MLP)-based model was introduced to predict the energy consumption of a smart building, including weather information. They compared the proposed model with linear regression (LR), support vector machines (SVM), gradient boosting machines (GBM), and RF methods. The experimental studies showed that the MLP model achieved the most successful result with values R2 of 64%, RMSE of 59.84%, mean absolute error (MAE) of 27.28%, and mean absolute percentage error (MAPE) of 27.09%. In a separate study, Shapi, Ramli & Awalin (2021) compared the performance of SVM, ANN, and k-nearest neighbors (KNN) using data collected from four tenants in a smart building in Malaysia. The findings indicated that each algorithm demonstrated varying performance levels based on RMSE and MAPE metrics for the tenants involved. Notably, SVM performed better than the other methods, achieving RMSE values of 4.75 and 3.59 for tenants A1 and A2, and with MAPE values of 19.09 and 43.97 for tenants B1 and B2. Priyadarshini et al. (2022a) proposed a reliable system for an IoT-based smart home environment using ML-based methods, including ARIMA, SARIMA, LSTM, Prophet, Light GBM, and vector autoregression (VAR). They performed time series analyses on the Internet of Things (IoT)-based smart home (IBSH) dataset and identified ARIMA as the best-performing method with an RMSE of 0.1806, MAE of 0.1491, and MSE of 0.0326 values. In a study by Moldovan & Slowik (2021), the authors explored the prediction of energy consumption for electrical appliances using various regression methods, including RF regression, ExtraTree (ET) regression, KNN regression, DT regression, and a hybrid approach called the multi-objective binary gray wolf optimization (MOBGWO-4ML). They utilized fourteen out of 28 features and found that the MOBGWO-4ML method performed the best, yielding the following values: MAPE of 0.349, RMSE of 0.568, R2 of 0.680, and MAE of 0.279. Ben (2021) performed a comparison study on electricity consumption prediction, applying random tree (RT), LR, SVM, and additive regression (AR) methods to the IBSH dataset. The results indicated that the RT method outperformed the other methods, with an MAE of 0.292, RMSE of 0.73, relative absolute error (RAE) of 50.62, and root relative squared error (RRSE) of 68.6. Priyadarshini et al. (2022b) presented an ensemble method based on DT, RF, and XGBoost for predicting electrical energy consumption. Their analysis, contrasting this ensemble method with DT, RF, XGBoost, and KNN techniques, demonstrated the superior performance of their approach. The error values for the first dataset were 0.000008 (MSE), 0.99999 (R2), 0.00072 (RMSE), and 0.00033 (MAE), while for the second dataset, the values were 0.000002 (MSE), 0.99999 (R2), 0.00078 (RMSE), and 0.00033 (MAE), respectively. In a study conducted by Mocanu et al. (2016), the authors examined the conditional restricted Boltzmann machine (CRBM) and the factored conditional restricted Boltzmann machine (FCRBM) models for time series prediction of energy consumption. They compared the performance of these models with ANN, SVM, and RNN methods using electrical power consumption data obtained from a residential customer. The results showed that the FCRBM method outperformed the other methods across different time periods. The reported RMSE values for FCRBM, ANN, SVM, RNN, and CRBM were 0.8995, 0.7971, 0.1702, 0.73, and 0.7971, respectively.

DL methods, which consist of multiple layers of neural networks, have become crucial for high-performance prediction in residential energy consumption due to their ability to model non-linear relationships. Kong et al. (2019) proposed an RNN model based on LSTM to address short-term load prediction for individual homes. This model was evaluated using a publicly accessible dataset of actual residential smart meter readings and demonstrated superior performance compared to traditional models such as backpropagation neural networks (BPNN), KNN, and extreme learning machines (ELM). The reported MAPE values for the LSTM model were 44.39%, 44.31%, and 44.06% at 2, 6, and 12-time steps, respectively. Das et al. (2020) carried out a study using the DL methods based on LSTM, Bi-LSTM, and GRU to predict occupant-specific miscellaneous electric loads. The performance of these models was tested on a range of plug-in loads associated with different devices. This investigation relied on data gathered from each sensor linked to the individual appliances. Findings indicated that both the Bi-LSTM and GRU models outperformed the LSTM model as the prediction interval extended. Cordeiro-Costas et al. (2023) undertook a comparative analysis to identify the most effective prediction model for predicting electricity consumption in single-family homes across the United States. The study compared ML methods, including RF, SVR, XGBoost, and MLP with DL-based methods like LSTM and 1-D convolutional neural networks (CONV-1D). Findings revealed that the LSTM method surpassed all other models, achieving a normalized Mean Bias Error (MBE) of −0.02% and a normalized RMSE of 2.76% on the validation dataset. For the test dataset, the LSTM model recorded a normalized MBE of −0.54% and a normalized RMSE of 4.74. Marino, Amarasinghe & Manic (2016) developed a novel method based on deep neural networks, especially LSTM for predicting energy load. They used two types of LSTM models: stand-alone LSTM and LSTM-based sequence-to-sequence (S2S) architecture. Both methods were tested on individual household electricity consumption data. The experimental results showed that the stand-alone LSTM did not perform well with 1-min resolution data, but it achieved satisfactory performance with 1-h resolution data. On the other hand, the LSTM-based S2S demonstrated successful performance for both resolutions, with RMSE values of 0.625 for the 1-h resolution dataset and 0.667 for the 1-min resolution dataset. Kim & Cho (2019a) proposed a DL-based model based on the autoencoder to predict energy demand in various situations. The model includes a projector that identifies a suitable state for a given situation and a predictor that estimates the energy demand from the defined state. In the experiments using the residential electricity power consumption data of 5 years, LR, DT, RF, MLP, LSTM, and stacking long short-term memory network (SLSTM) methods were utilized to evaluate the performance of the proposed model. The results obtained from experimental studies on datasets with different time resolutions of 15, 30, 45, and 60 min showed that the proposed model outperformed traditional models. This model obtained its highest performance from a 15-min periodic analysis, with an MSE of 0.2113, MAE of 0.2517 and MRE of 0.3625.

Hybrid DL approaches are gaining traction for predicting residential energy usage, thanks to their precision and the capacity to integrate various algorithm strengths. They exhibit adaptability to lifestyle data, versatility in managing time series, and have demonstrated effectiveness in a variety of contexts. Consequently, there has been a significant increase in recent research on residential energy consumption using these methods. Almalaq & Zhang (2019) proposed a new hybrid energy prediction approach combining the genetic algorithm (GA) and LSTM. This algorithm optimizes the objective function by considering the time window delays as well as the hidden neurons within the LSTM framework. The effectiveness of the GA-LSTM model was assessed for short-term predictions using data sourced from both residential and commercial buildings. The results obtained through a 10-fold cross-validation approach demonstrated that the proposed GA-LSTM method exhibited better performance than the traditional methods, achieving an RMSE of 0.213, coefficient of variation (CV) of 19.56%, and MAE of 0.074 for the first scenario. In parallel, for the second scenario, the GA-LSTM approach again exceeded the conventional methods with an RMSE of 0.43, CV of 8.38%, and MAE of 0.26. Kim & Cho (2019b) developed a hybrid CNN-LSTM neural network model that can effectively extract temporal and spatial features to predict residential energy consumption. They conducted experimental studies on four scenarios representing the individual household power consumption (IHPC) data arranged at minute, hourly, daily, and weekly resolutions. The performance comparison of the proposed method was made with LR and LSTM methods. The findings from all scenarios showed that the CNN-LSTM neural network model outperforms the LR and LSTM methods in terms of prediction accuracy. For the scenario with hourly resolution, the reported performance metrics for the proposed model were MSE = 0.3549, RMSE = 0.5957, MAE = 0.3317, and MAPE = 0.3283. Khan et al. (2020a) conducted a study on various DL-based prediction models and proposed a new hybrid model called CNN-LSTM-AE for predicting electricity consumption in residential and commercial buildings. The performance of the proposed model was compared with that of the CNN, LSTM, CNN-LSTM, and LSTM-AE models. The results of the study showed that in terms of the RMSE, MAE, and MSE metrics, the CNN-LSTM-AE model performed better than the other models for both residential and commercial buildings. For daily resolution data, the performance of the proposed hybrid model was summarized as follows: RMSE = 0.02, MAE = 0.01, and MSE = 0.0004 for residential buildings and RMSE = 0.01, MAE = 0.01, and MSE = 0.0003 for commercial buildings. Sajjad et al. (2020) proposed a hybrid model combining CNN and GRU DL methods to accurately detect prior electrical energy consumption in residential homes and tested this model on publicly available appliance energy prediction (AEP) and IHPC datasets. The performance of the proposed model was compared with that of the LR, DT, SVR, CNN, LSTM, and CNN-LSTM models. The results of the experimental studies demonstrated the applicability of the proposed hybrid CNN-GRU model more successfully in the real world than the basic models for residential homes. Specifically, for the IHEPC dataset, the reported performance results of the model were an RMSE of 0.47, MAE of 0.33, and MSE of 0.22. For the AEP dataset, the reported performance results were an RMSE of 0.31, MAE of 0.24, and MSE of 0.09. Zang et al. (2021) developed a novel day-ahead residential load prediction method that incorporated feature engineering, pooling, and a hybrid DL model. The hybrid model combined LSTM with a self-attention mechanism (SAM). The model was verified using a practical dataset that included multiple residential users. The experimental results demonstrate that the performance of the proposed method varies depending on the data pool from different groups of users. The best performance was achieved using a four-user data pool with 49-time steps and 24 feature sizes. The performance of the model, in terms of MAPE, MAE, and RMSE, was reported to be 15.33%, 56.86, and 82.50 kW, respectively. Fang et al. (2021) proposed a hybrid deep transfer learning strategy that combines LSTM and a domain-adversarial neural network (DANN) to address the issue of limited data in training models for building energy prediction. They conducted experimental studies by using different models to verify the performance of the proposed method. The results showed that the proposed strategy significantly improves building energy prediction compared with models trained solely on target, source, or both target and source data without transfer learning. Syed et al. (2021) developed a hybrid DL model that combines the advantages of LSTM, BI-LSTM, and RNN for energy consumption prediction accuracy in smart buildings. They compared the proposed model with other commonly used hybrid models, including CNN-LSTM, ConvLSTM, LSTM encoder-decoder model, and SLSTM. The evaluation was conducted on two real energy consumption datasets from smart buildings. The results revealed that the proposed model outperformed the LSTM-based hybrid models, yielding MAPE values of 2.00% in the first case study and 3.71% in the first and second case studies, respectively. Moreover, for the multi-step week-ahead daily forecasting, the proposed model has an improvement of 8.368% and 20.99% in MAPE compared with the LSTM-based model.

On the other hand, ensemble DL methods, as in hybrid methods, have recently become more prominent in residential energy consumption prediction than other methods due to their improved prediction accuracy as a result of combining the strengths of different methods. Khan et al. (2021a) presented an ensemble approach that combined LSTM and KF methods to predict short-term energy consumption in a multifamily residential building in South Korea. The proposed model was compared with the stand-alone LSTM and KF methods and successful ML-based methods such as RF, AdaBoost, XGBoost, and GB. The results demonstrate that the proposed ensemble model performed better than the other methods, with an MAE of 373.580, RMSE of 487, MAPE of 3.264, and R2 of 0.966. Khan et al. (2021b) developed a spatial and temporal ensemble model for the prediction of short-term electricity consumption. This model incorporates two deep learning models, LSTM and GRU, and employs cluster analysis with the k-means algorithm to discover apartment-level electricity consumption profiles. The proposed model was tested using high-resolution electricity consumption data from smart meters at both building and floor levels. Its effectiveness in predicting energy consumption was compared with those of widely used ML and DL methods. The experimental results confirmed that the model successfully captured the sequential behavior of electricity consumption, demonstrating superior building and floor-level prediction performance, with the lowest MAPE of 4.182 and 4.54, respectively. Hadjout et al. (2022) proposed an ensemble DL method that combines LSTM, GRU, and TCN models to accurately predict long-term electricity consumption for the Algerian economic sector. The researchers used the weighted average technique to combine the outputs of DL models, and the weight coefficients were obtained using a grid search (GS) algorithm. The study, which utilized the Bejaia High Voltage Type A (HVA) consumer energy consumption dataset, revealed that the proposed ensemble model performed sufficiently well to meet the company’s requirements and better than the prediction of stand-alone models. The statistical significance of the model was evaluated using the Wilcoxon signed-rank test, which showed a p-value less than 0.05, for all paired combinations. Sujan Reddy et al. (2022) created two stacking ensemble models for electrical energy consumption prediction, combining LSTM, EvTree, RF and NN basic predictor outputs through XGBoost and GBM, and they tested these models on a standard dataset containing 500,000 energy consumption data. The results obtained from their study showed that the stacking model with the XGBoost combiner reduced the training time of the second layer by a factor of approximately 10 and improved the RMSE by 39%. Guo et al. (2024) developed a novel stacking ensemble method that combines the outputs of ANN, XGBoost, LSTM, Stacked LSTM, and BI-LSTM models through a Lasso regressor. They applied this method to two real-world datasets and achieved better accuracy with MAPE values of 5.99% and 7.80%, respectively, compared to similar methods. Irfan et al. (2023) proposed an ensemble model combining the RNN and LSTM methods to effectively predict multi-region hourly power usage. The performance of this model was evaluated using a dataset collected from thirteen different regions between 2004 and 2018. The results obtained from the experimental studies demonstrated that the model performed better than other base methods. Some statistical results, such as the RMSE, R2, and MAPE for the proposed model, were reported as 0.1658, 0.9731, and 1.9238, respectively.

The literature review results discussed above show that most AI-based methods, including adaptive, ML, DL, hybrid DL, and ensemble DL methods, can produce successful prediction results on the specific datasets to which they are applied. Among them, hybrid and ensemble DL methods outperform standalone and adaptive methods. Therefore, hybrid methods with an adjustable parameter structure and a combination of various DL-based methods have been widely used in numerous energy consumption prediction studies. Ensemble DL methods that combine the different strengths of multiple DL models have gained popularity recently due to the high-performance advantages they provide in this area. Average or weighted average techniques have been used in a limited number of studies to combine the outputs of base DL predictors owing to performance issues from their linear structures arising from fixed weights. Instead, stacking ensemble DL methods that use ML techniques as meta-combiners capable of successfully modeling variations in the outputs of base predictors are widely preferred due to their high-performance effects. However, along with the long training times of ML combiners, overfitting and generalization issues in high-density datasets can often lead to performance drops across the entire model. Accordingly, the need for a successful prediction process is to introduce a structurally simpler and functionally more effective meta-combiner that can minimize these issues. Based on this requirement, this study proposes a novel stacking ensemble DL method. It combines the output of five DL-based methods proven to be effective in time-series problems through a simplified ANFIS architecture with the best score transfer as a meta-combiner.

Materials and Methods

This section outlines the materials and methodologies used in this study. The researchers first provided a detailed explanation of the datasets used for energy consumption prediction and their characteristics. They then thoroughly described the proposed deep learning-based novel ensemble (DLBNE) method with best score transferred-adaptive neuro fuzzy inference system (BSTANFIS), including its construction, constituent units, and the roles of these components. Subsequently, the researchers presented the ML and DLBWA ensemble methods, which were used for comparison with the proposed DLBNE approach. Finally, an overview of the performance metrics applied to assess the effectiveness of the different models was presented.

Dataset

The method proposed in this study incorporates DL algorithms to capture the spatial relationships among different features and to learn long- and short-term dependencies. Accordingly, the selection of the dataset focused on two key aspects to effectively showcase the performance of these algorithms: (1) a comprehensive feature map combining various features and (2) sufficient data density to improve generalization ability. In addition, it is important to consider that the dataset contains real-world data when selecting the dataset because it is crucial to test the performance of the proposed method in practical applications. In this context, the IBSH and IHPC datasets, including real-world data on household energy consumption, and the homestead city electricity consumption (HCEC) dataset combining different features on city energy consumption were used in the experimental studies.

IoT-based smart home (IBSH) dataset

The IBSH dataset, available on Kaggle website (Singh, 2018), contains the power consumption data in Kilowatts and weather information measured at 1-min intervals over a span of 350 days. The dataset has 5,003,911 data points in total and is made up of 32 features. A total of 18 of these features represent power consumption, 13 contain weather information, while one feature includes time information, as shown in Table 1.

Table 1 Feature information for the IBSH dataset.

Feature labels	
Time	Microwave	Fridge	Pressure	
Use	Living room	Wine cellar	windSpeed	
Gen	Solar	Garage door	cloudCover	
House overall	Weather	Kitchen 12	windBearing	
Dishwasher	Temperature	Kitchen 14	dewPoint	
Furnace 1	Humidity	Kitchen 38	precipProbability	
Furnace 2	Visibility	Barn	precipIntensity	
Home Office	apparentTemperature	Well	Summary	

Each column in the dataset represents the value changes labeled with the features in Table 1. The data contained in the columns, apart from the house overall column, serves as inputs for energy consumption prediction methods. The house overall column, which gives the total power consumed by all devices, is considered as the real data class to be used in training and testing processes.

Individual household power consumption (IHPC) dataset

This dataset, available on UCI Machine Learning Repository (Hebrail & Berard, 2006), consists of power consumption data collected from a single house in Sceaux, France, over a 47-month period from December 2006 to November 2010, measured at 1-min intervals. Although 25,979 measurements are missing from the 2,075,259 measurements in the dataset, this data can be easily addressed during the pre-processing phase to prepare it for prediction. In addition to date and time features, the dataset includes variables related to the global minute-averaged active power (in kilowatt), the global minute-averaged reactive power (in kilowatt), the minute-averaged voltage (in volt), and the minute-averaged current intensity (in ampere). It also covers variables relevant to the active energy consumption in watt-hours obtained from three sub-meters (Sub-metering-1, Sub-metering-2, Sub-metering-3) in various areas of the house. The active energy consumption measurements include the first, second, and third sub-meters used for kitchen, laundry, and climate control systems. A new feature representing total energy consumption per minute (TEC) was later added to the dataset in addition to the existing features to be used as a target variable in the training and testing phases. The equation for obtaining this feature is as follows.

(1) TEC=Globalactivepower∗100060−Submetering1−Submetering2−Submetering3.

Homestead city electricity consumption (HCEC) dataset

The HCEC data is a publicly available dataset on Kaggle website (Jain, 2020) involving hourly electric energy consumption from Homestead city in the United States. This dataset contains 22,201 data with eighteen variables serving as features. Among these variables, two represent date and time, one denotes the target energy consumption, while the remaining ones include external factors related to weather information such as temperature, humidity, rainfall, and wind speed.

Deep learning-based novel ensemble method with best score transferred-ANFIS

The DLBNE method proposed in this study incorporates the strengths of DL-based methods with the adaptive and fuzzy logic capabilities of ANFIS to achieve high-performance energy consumption prediction. As this method combines the prediction scores of base DL predictors through a meta-combiner, it represents a novel stacking-type ensemble method. The choice of base predictors for the method is based on the structure of the data to be used. Electricity energy consumption data are complex time series containing both short- and long-term sequential dependencies. Therefore, the researchers leverage five well-established DL models that have proven successful in this area: RNN, LSTM, BI-LSTM, GRU, and CNN. The RNN model is employed to capture the short-term sequential dependencies in the data. LSTM, BI-LSTM, and GRU models are used to effectively model long-term sequential dependencies. Additionally, the CNN model is incorporated to identify local patterns and detect anomalies in the electricity consumption data. The most distinguishing feature of the proposed method apart from existing stacking ensemble DL methods is its use of a meta-combiner that merges the outputs of base predictors through the best score transfer approach with the ANFIS architecture. The ability of ANFIS to adaptively tune its parameters based on the problem and generate new inferences from historical data, similar to human reasoning, makes it an effective component for combining the nonlinear scores of DL predictors in the method. The best score transfer approach, on the other hand, simplifies ANFIS structurally by reducing the number of inputs. This structure brings with it two important contributions: it shortens the training time and minimizes the performance issues arising from overfitting in the base predictors. Consequently, the ANFIS architecture with the best score transfer becomes a more robust and effective combiner than other meta-combiners used in stacking ensemble methods in terms of performance.

Figure 1 illustrates the schematic diagram of the DLBNE method with BST-ANFIS, which is proposed to create a robust and efficient model for energy consumption prediction. As seen from the schematic diagram, it is structurally composed of seven units. Five of these units perform the main functions, while the remaining two measure the method’s performance. In this structure, the data preprocessor, five-fold CV data splitter, base predictors, input decision-maker, and prediction combiner carry out the main functions of the proposed method, while the model evaluator and mean evaluator are used to assess its performance. The data preprocessor enhances data quality through cleaning, fitting, and normalization processes. The five-fold CV data splitter obtains cross-validation datasets that include training and testing data to verify the accuracy of the proposed method. The base predictor individually determines the prediction scores of DL-based methods using data from the five-fold CV data splitter. The input decision-maker executes an approach that identifies the best scores to be transferred from the base predictors to the prediction combiner (ANFIS). As a result, the two best-scoring inputs to be sent to the prediction combiner are obtained through this approach, utilizing the predictor selector and B-nearest scorer procedures. The prediction combiner synthesizes these scores through ANFIS, resulting in the final prediction output. Among the last two units used for performance evaluation, the model evaluator assesses the performance of the method for each fold, while the mean evaluator computes the overall performance based on the findings of the five-fold performance.

Figure 1 The schematic diagram of the DLBNE method with BST-ANFIS.

The operation of the DLBNE method with BST-ANFIS can be explained in three phases: training phase I, training phase II, and testing phase. In training phase I, also referred to as the preparation phase, the base predictors are trained using active training data sourced from the five-fold CV data splitter. During this process, the prediction scores for this data, along with their corresponding real labels, are recorded in the training data score database. At the end of this phase, the DL predictor that demonstrates the lowest error value—assessed through the MAE of the base predictors—is selected as the first best-scoring input to be applied to the prediction combiner. The training phase II focuses on the ANFIS training process. Here, the data stored in the training data scores database serves as training input. The ANFIS training utilizes the output score from the selected DL predictor identified in Training Phase I as its first input. The second input is derived from the output of the DL predictor that has the lowest absolute error associated with the first input at each sampling loop. The final phase, the testing phase, involves obtaining predictions for energy consumption and evaluating performance results on test data provided by the five-fold CV data splitter. Initially, active fold test data is applied to the trained DL models in the base predictor. The resulting scores from these base predictors for each sampling loop are then transferred to the input decision-maker. Similar to Training Phase II, this unit identifies the two best-scoring inputs for ANFIS by applying MAE and absolute error (AE) criteria to the scores produced by the DL predictors for the test data. These selected inputs are dynamically fed into ANFIS during each sampling loop, culminating in a combined score that represents the final energy consumption prediction. After completing one sampling loop, the split data index (Splord) is incremented, and identical operations are repeated for subsequent fold data across all phases. When this index reaches five, the average performance for the DLBNE method with BST-ANFIS is calculated using the mean evaluator. The details about the proposed method are described in the following subheadings.

Data preprocessor

The data preprocessor is a critical unit used to eliminate the negative impacts of missing, incorrect, and inconsistent values in the datasets. This ensures the data can be easily understood and interpreted by predictive models. In this context, the preprocessor performed various data preprocessing tasks such as data cleaning, organization, removal, and normalization on the datasets. The study utilized three datasets-HCEC with hourly resolution, and IBSH and IHPC with minute-resolution. To harmonize the resolutions, the preprocessor first rearranged the IBSH and IHPC datasets into hourly format to match the HCEC dataset. The next processing steps of the preprocessor can be summarized as follows for each dataset: For the IBSH dataset, the preprocessor created two new columns by merging related features (Furnace 1 & 2, Kitchen 14 & 38). It also removed the “kW” unit from column names and deleted string-type columns. Invalid “cloudCover” values were replaced with the next valid entry, and the time column was converted to a standardized datetime (Y-m-d H-M-S) format. Additionally, one of the same highly correlated columns (use, house overall, gen Solar) were pruned to avoid redundancy. In the IHPC dataset, the date and time columns were combined into a single time column. Missing values were imputed using the column averages. In the HCEC dataset, the index column was deleted. The “Homestead” prefix was also removed from column names, and the missing values were filled with the average value of the respective column. Finally, the preprocessor completed the process by normalizing all column values across the datasets.

5-fold CV data splitter

The five-fold CV data splitter is a unit that helps to check the effectiveness and reliability of the DLBNE method with BST-ANFIS across different datasets. It applies the k-fold cross-validation technique to create the training and testing datasets. Several key parameters govern the process. First, the k parameter was set to five, meaning that the cross-validation used five folds or subsets of the data. The second parameter, called random seed, was set to 42. This ensures that the dataset is randomly divided into folds in a consistent manner during the run of the algorithm, which helps reproducibility. In addition, there is a shuffle parameter set to “True,” meaning that the data will be randomized before starting the cross-validation, reducing any bias from the original order of the data. Using these parameters, the five-fold CV data splitter shuffled the data, split it into five folds, and then performed the k-fold cross-validation process. Each fold takes turns as the testing data, whereas the others are used for training the model. This was repeated for each fold, and at the end of the process, the unit created CV datasets with five training and testing sets.

Base predictors

This unit individually obtains the scores of DL-based methods for each CV dataset. It includes five DL-based methods: CNN, RNN, LSTM, Bi-LSTM, and GRU. The basic descriptions of these methods are presented below in subheadings.

RNN: It is an artificial neural network-based method that can learn patterns and relationships in sequential data by utilizing historical information. This method stands out from other neural networks because of its hidden states, which serve as memory components, preserving some of the information from the sequence that has been processed up to that point (Song et al., 2023). The RNN continuously updates its hidden states with each new input and preceding hidden states, making decisions that depend on both. This feature makes it an effective method for solving problems in natural language processing, anomaly detection, text and sentence generation, and time series analysis. As illustrated in Fig. 2A, RNN cells are typically shown as neural network nodes with a loop, symbolizing the transfer of information from one sequence step to the next. Each cell, indicative of the hidden states, operates as a function that merges the current input with prior hidden states to produce the output, which is then derived by processing the relevant state information through a function. Given this information, the relationship between the input and output of an RNN cell can be described by Eqs. (2)–(6). The following equations represent the input ( xt), network weights and thresholds ( wt, vt and, nt), aggregation information ( at), hidden state information ( ht), and output ( yt).

Figure 2 The schematics of the DL-based four base predictors: (A) RNN cell, (B) LSTM cell, (C) Bi-LSTM and (D) GRU cell.

(2) x→x1,x2,......xt

(3) w1,…,wt,v1,…,vtandni,…,nt

(4) at=vt−1.ht−1+wt.xt+bt

(5) ht=σ(at)

(6) yt=f(ht).

Sigmoid and ReLU are the functions most frequently utilized for the hidden state (σ(.)), while SoftMax is used for output (f(.)). The symbol ⊙ denotes pointwise multiplication. Training RNNs with the backpropagation algorithm results in the vanishing gradient problem, similar to that in conventional neural networks. Consequently, this issue hinders these networks’ ability to grasp and learn long-term dependencies present in sequential data (Jozefowicz, Zaremba & Sutskever, 2015).

LSTM: It is a recurrent deep network architecture specifically designed to address the vanishing gradient problem in RNNs (Sunjaya, Permai & Gunawan, 2023). These networks excel in learning long-term dependencies, which standard RNNs often struggle with, thereby providing efficient solutions for complex time series data. In general, an LSTM layer in an LSTM network comprises a series of memory blocks that are interconnected in a recurrent manner. Each contains memory cells that are also recurrently connected, along with four key components, as depicted in Fig. 2B: input gate ( it), forget gate ( ft), output gate ( ot), and cell state ( ct). In this structure, the input gate contributes significantly to updating the cell state by determining which new pieces of information will be stored. The forget gate determines which information from the previous steps in the sequence should be discarded or retained depending on the forget vector. The output gate is responsible for defining the next hidden state and identifying information that will come from the cell state (Fayaz et al., 2023). The cell state is a communication line that carries meaningful information across cells to make predictions. Accordingly, the basic equations defining the information flow in an LSTM cell are as follows.

(7) it=σ(Wixt+Uiht−1+bi)

(8) ft=σ(Wfxt+Ufh(t−1)+bf)

(9) c~t=tanh(Wcxt+Ucht−1+bc)

(10) ct=ft⊙c(t−1)+it⊙c~t

(11) ot=σ(Woxt+Uoh(t−1)+bo)

(12) ht=ot⊙tanh(ct).

Here Wi, Wf, Wc,Wo are the weight matrices for the input gate, Ui, Uf, Uc,Uo are the weight matrices for the hidden state, and bi, bf, bc,bo are the bias terms for the input gate, forget gate, output gate, and cell state operations, respectively. The symbols σ and tanh correspond to the sigmoid and hyperbolic tangent activation functions.

Bi-LSTM: The method developed by Schuster & Paliwal (1997) is a specialized version of the LSTM network. BI-LSTM is a robust approach for capturing temporal patterns and generalizing unseen data. Its popularity has surged because it effectively addresses many ML problems. We can see its successful use in various fields, such as natural language processing, speech recognition, audio classification, and time series prediction. As shown in Fig. 2C, it consists of two LSTM layers that allow the input to flow forward and backward (Xia et al., 2020). This setup enables it to obtain model output by evaluating historical and future information. Thus, the network can better understand the input sequence by capturing both contextual information. Furthermore, it solves the gradient and information loss issues that often arise in traditional RNN layers during training (Wan et al., 2022). In Bi-LSTM, the final output ( yt) is obtained by combining the forward LSTM hidden state ( h→t) and backward LSTM hidden state ( h←t) based on the input data sequence ( xt). Equations (13)–(15) give the formulas for calculating h→t, h←t, and yt outputs.

(13) h→t=LSTM(xt,h→t−1)

(14) h←t=LSTM(xt,h←t+1)

(15) yt=Wh→yh→t+Wh↼yh↼t+by.

Here, LSTM (•) symbolizes the LSTM equations given in Eqs. (7)–(12), Wh→y corresponds to the forward LSTM weights, Wh↼y represents the backward LSTM weights, and by defines the output layer bias of the Bi-LSTM.

GRU: The method proposed by Cho et al. (2014) is another variation of recurrent networks endowed with memory capabilities. GRU, like LSTM networks, possesses gated mechanisms that enable it to filter information, thereby allowing the retention of significant data in long-term memory for subsequent retrieval, as needed. It employs fewer parameters and features a more streamlined architecture than LSTM, which can result in expedited training times. However, a reduced number of parameters does not always guarantee a better performance. The decision between the GRU and LSTM is contingent upon the specific task at hand and the dataset utilized. As illustrated in Fig. 2D, a GRU cell incorporates two gating mechanisms that offer a flexible approach to regulate the information flow in the network: the reset gate and the update gate. Depending on the input data, the reset gate determines how much past information should be discarded, thus enabling the dynamic selection of pertinent details from previous states for current processing. The update gate regulates the preservation and transfer of prior information to the next time step, striking a balance between new inputs and existing memory. Notably, the GRU lacks an additional memory cell for storing information; instead, it solely governs data in its cell state unit (Gao et al., 2020; Tang et al., 2023). The mathematical equations corresponding to GRU cell are as follows.

(16) rt=σ(Wrxt+Vrh(t−1)+br)

(17) zt=σ(Wzxt+Vzh(t−1)+bz)

(18) c~t=tanh(Wcxt+Vc(rt⊙h(t−1))+bc)

(19) ct=(1−zt)⊙c(t−1)+zt⊙c~t

(20) ht=ct.

In the abovementioned equations, zt is the update gate, rt is the reset gate, and ht is the hidden state. The symbols used in the equations are as follows: Wr, Wz, and Vr,Vz represent weight matrices, while br and bz are bias vectors. In addition, the symbols σ and tanh represent the sigmoid activation function and the hyperbolic tangent activation function.

CNN: The method proposed by Lecun, Bengio & Hinton (2015) is a deep feedforward neural network that is crucial in extracting spatial features for time series data, image recognition, and classification problems. Just like how a neuron in the brain processes information and sends it throughout the body, the artificial neurons in CNNs do the same with inputs and produce outputs (Happiest Minds, 2023). Figure 3 illustrates the architecture of a fundamental CNN. It comprises a convolutional layer, pooling layer, flatten layer, and fully connected layer. In this structure, the convolutional layer extracts local features from the input data using filters, also known as kernels. Each filter is designed to identify and capture specific features or patterns in these data. The pooling layer then reduces the dimensions of the feature map created by the convolutional layer, keeping the essential information. Maximum pooling is the most common technique used here; it picks the highest value in a feature map as the most important feature for extraction. The flatten layer serves as an indispensable element that enhances overall neural network functionality; it transforms multidimensional data from both convolutional and pooling layers into a one-dimensional vector before relaying results to fully connected layers. These fully connected layers then process these feature vectors to allocate specific output values corresponding to input data. Finally, these values go through functions like Sigmoid and SoftMax to create probability distributions, helping to determine the class of the input data (Zheng et al., 2023).

Figure 3 The architecture of a fundamental CNN.

The output of CNN ( fth) is mathematically represented as given below.

(21) fth=∑i=1p⁡∑k=1q⁡Wik×Xik.

In the aforementioned equation, Xik defines the input sequence, and Wik denotes the kernel weight. The symbols p, q, and h represent the length of the input sequence, the feature dimension of the instances, and the convolutional kernel, respectively. The representation of the convolutional layers in the CNN architecture is expressed through Eqs. (22) and (23).

(22) uikv=f((Wv∗x)ik+bv)

(23) f(x)=max(0,x).

Here, Eq. (22) is the activation function, and Eq. (23) is the ReLU function. The symbol Wv in the activation function represents the kernel weight.

Input decision-maker

The input decision maker unit is responsible for selecting the two best-scoring inputs from the outputs of the base predictors, which are then forwarded to the prediction combiner (ANFIS), all in pursuit of producing a high-performance prediction result. This unit accomplishes its objective through two fundamental procedures: a predictor selector and a B-nearest scorer. The predictor selector procedure determines the first best-scoring input to be sent to the prediction combiner, while the B-nearest scorer procedure finds the second best-scoring input.

In this process, the data array derived from the output of the base predictor unit during training phase I is categorized into two distinct groups: real data array ( Rtrn) and predicted data array ( BPtrn). In order to select the first best-scoring input, the predictor selector first applies the MAE criterion, as described in Eq. (24), to these data arrays to calculate the average error score for each DL predictor.

(24) MAEtrni(Rtrn,BPtrn)=∑j=1m⁡|Rtrn(j)−BPtrni(j)|mi=1,2,…5.

Here, Rtrn(j) is the jth value of the real data array for training phase I, BPtrni(j) is the jth prediction score of the ith predictor for training phase I, and m is the total number of instances for the same phase. Upon obtaining the MAE measurements, the predictor selector identifies the kth DL predictor that produces the lowest measurement using Eq. (25) and assigns it as the first best-scoring input ( IANF(1)) to the predictor combiner for both training phase II and testing phase.

(25) IANF(1)=Bestprd(BPtrn,MAEtrn)={BPtrnimin(MAEtrni)}i=k.

In the IANF(1) equation, Bestprd() is the function that finds the first best-scoring input based on the prediction scores of DL predictors for the training phase I and the MAE measurements for these predictors ( MAEtrn), and BPtrnk is the scores of the best predictor produced by this function. In order to determine the second best-scoring input for the prediction combiner, the B-nearest scorer performs a procedure similar to that of the predictor selector, which chooses the first best-scoring input. The differences between them lie in the operating phases, the criteria used to determine the best score, and the data source referenced when applying this criterion. In fact, this input corresponds to the prediction result that is closest to the DL predictor representing the first best-scoring input, dynamically in each sampling loop of training II and testing phases. Accordingly, in each sampling loop, the B-nearest scorer first defines the score of the DL predictor representing the first input of the prediction combiner as a reference for determining the score to be assigned as the second best-scoring input to this unit. Then, it obtains the absolute errors (AEs) of the other DL predictors with respect to this reference, using Eq. (26).

(26) AEn(Bestprd,BP)=|Bestprd(j)−BPn(j)n≠k|

Here, n denotes the index number of the predictors outside the best predictor k. AEjn represents the absolute error of the nth predictor in the jth sampling loop, while Bestprd(j) refers to the score of the DL predictor assigned as the first best-scoring input to the prediction combiner in the jth sampling loop. Finally, the B-nearest scorer assigns the DL predictor score with the lowest AE measurement, derived from Eq. (27), as the second best-scoring input to the prediction combiner.

(27) IANF(2)=Nearestprd(BP,AE)={BPn(j)min(AEn(j))}n=p.

Here, the output scores of various predictors can be allocated to the second best-scoring input of the prediction combiner in each sampling loop based on AE measurements. Consequently, this input demonstrates dynamic behavior, with its value changing throughout the process.

Prediction combiner

In the proposed method, the prediction combiner obtains the final result by combining two best-scoring predictor outputs that come to its input. The literature review results show that, because of the dynamic structure of electrical energy consumption data from different sources, average and weighted average techniques fail to provide the desired performance in this process. Instead, ML-based meta-combiners, such as XGBoost, GBM, SVM, and KNN, have been successfully applied in many studies. However, these combiners can face several performance issues, particularly in high-dimensional datasets, owing to overfitting and generalization problems. The fact that ANFIS architectures combine fuzzy inference systems, which are known to be effective in modeling uncertainties, with adaptive neural networks that optimize their own parameters, can make it an effective combiner for solving these problems. In particular, a simplified ANFIS configuration with fewer rules and a hybrid learning mechanism can perform better in the combination process by eliminating overfitting and generalization issues.

Two types of ANFIS architectures have been proposed in the literature: Type-1 ANFIS and Type-2 ANFIS. One of them, Type-2 ANFIS, is an architecture proposed to better manage the uncertainties caused by fuzzy sets in the Type-1 ANFIS, which is also commonly referred to as ANFIS (Dabiri Aghdam et al., 2022; Alberto-Rodríguez, López-Morales & Ramos-Fernández, 2024). The ability to model uncertainties appears to be an advantage of this architecture in terms of performance; however, the computational complexity and additional resource requirements in terms of training the model constitute the most significant limitations in practice (Chen et al., 2017). Therefore, Type-1 ANFIS is more widely used than Type-2 ANFIS in current applications. In this study, the proposed method utilizes ANFIS (Type-1 ANFIS) as the prediction combiner. This choice was made to avoid further increasing the computational burden imposed by the DL models, and also due to the successful performance and relatively low computational complexity of Type-1 ANFIS for this type of problem. Figure 4 shows the ANFIS architecture used as a two-input and one-output prediction combiner in the proposed method.

Figure 4 The ANFIS architecture used in the proposed method as a two-input and one-output prediction combiner.

Determining the most appropriate ANFIS configuration for the proposed method is crucial for reducing complexity and enhancing model performance. In this regard, we focused on specifying the parameters that affect the ANFIS configuration, namely, the number of inputs, sets used for each input and type of membership function (MF). The selection of the shape or type of MF is specific to the characteristics of the problem. Therefore, in most application studies, the type of MF is determined through trial-and-error experiments. Triangular and Gaussian MFs are commonly preferred in literature reviews examining these processes. Triangular MF provides an ideal solution for systems requiring simple and fast solutions, while Gaussian MF offers more stable results for complex, nonlinear, and noisy systems (Wu, 2012; Sadollah, 2018). Similar to the studies in the literature, the parameters affecting the ANFIS configuration in this study were determined through experimental trials. Under 0.7 data splitting conditions, experiments were conducted on sampling data from the IBSH dataset by varying the number of inputs from two to four and the number of membership functions from three to nine, using triangular, trapezoidal, Gaussian, and Gaussian2 MFs. The evaluations based on the RMSE performance and structural simplicity indicated that the ANFIS architecture with two inputs, nine membership sets, 81 rules, and Gaussian MFs, achieving an RMSE of 0.000675278, was the most suitable configuration for the proposed method. This architecture is structured into five layers, as depicted in Fig. 4.

Layer 1, also referred to as the fuzzification layer, verbalizes the input information and computes the membership degree for associated linguistic labels. The rectangles in this layer represent the adaptive nodes. Each node is verbally labeled, and its defined membership function decides the membership degree of the input data with that particular label. The equation that determines the output of a node defined by Gaussian membership functions in this layer is as follows:

MFj(IANF(i))=fj(IANF(i);σ,c)=e−(IANF(i)−cj)22σj2i=1,2j=1,2…9

(28) Oij1=MFj(IANF(i)).

Here, Oij1 defines the output of the jth node for the ith input, and MFj(IANF(i)) denotes the jth membership function for the ith input. cj and σj are the formal parameters that shape the jth membership function.

Layer 2 consists of constant nodes labeled with π. These nodes calculate the firing strengths of rules in the ANFIS. The firing strengths represent how much each rule contributes to the overall system output. The node outputs are determined by multiplying the pairwise membership values from the previous layer. Alternatively, the AND operator can also be used instead of the multiplication operation to compute the node outputs.

(29) Ok2=wk=MFj(IANF(1))×MFj(IANF(2))j=1,2,…,9k=1,2,…..,8.

Here, the Ok2 is the output of the kth node in Layer 2, and wk is the firing strength of the kth node.

Layer 3 normalizes the firing strengths from Layer 2 using constant nodes labeled with N. Therefore, this layer is also referred to as the normalization layer. The output of the kth node in the normalization layer is calculated as follows.

(30) Ok3=w^k=wk∑k=181⁡wk.

Here, Ok3 represents the output of the kth node in Layer 3, while w^k denotes the normalized firing strength for the kth node in the same layer. The denominator ∑k=181⁡wk indicates the total sum of firing strengths across the 81 nodes in Layer 2.

Layer 4, also known as the defuzzification layer, involves multiplying the normalized firing strengths from Layer 3 with the rule outputs represented by a first-order linear polynomial equation. The rule structure for the proposed prediction combiner ANFIS is given below.

(31) R1:IFIANF(1)=F1veIANF(2)=F1THENf1=p1×IANF(1)+q1×IANF(2)+r1R2:IFIANF(1)=F1veIANF(2)=F2THENf2=p2×IANF(1)+q1×IANF(2)+r2⋯⋯⋯⋯⋯⋯⋯⋯⋯⋯⋯⋯⋯⋯⋯⋯⋯⋯⋯⋯⋯⋯⋯⋯.R81:IFIANF(1)=F9veIANF(2)=F9THENf81=p81×IANF(1)+q81×IANF(2)+r81.

The symbols F1 to F9 represent the antecedent parameters of the rules for inputs, while those f1 to f81 denote the consequent parameters of the first-order polynomial rules. The output of the kth adaptive node ( Ok4), which determines the contribution of the rules in this layer to the total output, is obtained by Eq. (32).

(32) Ok4=w^k×fk=w^k×(pk×IANF(1)+qk×IANF(2)+rk).

Here, the variables pk, qk and rk are the set of consequent parameters for the kth rule.

Layer 5, called also as the output layer, consists of a single node marked with the ∑ symbol. This node sums up all the rule outputs from Layer 4 to produce the final ANFIS output. The process can be mathematically expressed as follows.

(33) O5=∑k=181⁡w^k×fk=∑k=181⁡wk×fk∑k=181⁡wk.

The expression ∑k=181⁡w^k×fk in the equation represents the sum of the 81 defuzzified rule outputs from Layer 4.

The operation of ANFIS can be explained in two separate phases: training and testing. During the training phase, the input (antecedent) and output (consequent) parameters of the fuzzy rules are adjusted using backpropagation or a hybrid learning algorithm (Zhou, Herrera-Herbert & Hidalgo, 2017). This study employs a hybrid learning algorithm, enabling the ANFIS network to effectively update its parameters. The algorithm manages the training process using two well-known methods: least-squares estimation (LSE) and gradient descent (GD). During the forward pass, the LSE method is used to fine-tune the output (consequent) parameters, while keeping the input (antecedent) parameters constant. This is achieved by minimizing the sum of squared differences between the target and predicted outputs. In the backward pass, the GD method is applied to adjust the input (antecedent) parameters, maintaining the output (consequent) parameters fixed. This method computes the derivatives of the error function with respect to the antecedent parameters and updates them accordingly to minimize the error. After the training phase is complete, the testing phase determines the final output by applying test data to the trained ANFIS model.

Model evaluator

Evaluating the accuracy of regression models is crucial in this process. In the proposed method, the model evaluator serves as the unit responsible for evaluating the model’s accuracy for each fold of data. It performs this evaluation using well-recognized metrics from literature, including MAE, RMSE, MSE, and MAPE. Table 2 shows the basic definitions and formulas for these metrics.

Table 2 The basic definitions and formulas for performance metrics.

Performance metrics	Formula	Definition	
MSE	MSE=∑i=1n⁡(yi−y˙i)2n	It gives the mean squared difference between the real value and the predicted value. In general, this metric is considered as a measure of the proximity of a fitted line to the data points.	
RMSE	RMSE=∑i=1n⁡(yi−y˙i)2n	It measures how far predictions are from the real values and its value is calculated by taking the square root of the mean of the squared errors.	
MAE	MAE=∑i=1n⁡|yi−y˙i|n	It provides the mean absolute deviation between predicted values and real values.	
MAPE	MAPE=∑i=1n|yi−y˙i/yi|n	It represents the mean of the absolute percentage differences between predictions and real values.	
Note:

n, The total number of testing samples; yi, the real value of testing samples; y˙i, the prediction value of testing samples.

The RMSE metric is particularly useful in cases where discrepancies between predictions are substantial. Similarly, the MSE metric is concerned with the magnitude of errors; however, its interpretation can be more complex than that of the RMSE due to its reliance on the squares of errors. MAPE metric deals with percentage error and facilitates comparison between different datasets (Brownlee, 2021a; Agrawal, 2023). Finally, the MAE metric is frequently employed in cases where working with datasets containing outliers is crucial and assessing the absolute error in predictions is of primary importance.

Mean evaluator

The mean evaluator is a unit that computes the average performance of the proposed method alongside others based on the performances measured for each fold of the data. It determines the final performance of all methods used in energy consumption prediction by averaging the individual performances obtained for the five-fold CV dataset. Consequently, the mean evaluator’s output for the kth performance metric is found by Eq. (34).

(34) Pavrgk=15×∑i=15⁡Pikk=1,2⋯4.

Here Pavrgk gives the final performance of the kth metric based on the five-fold CV technique, while Pik provides the individual performance of the kth metric for the ith fold of the data.

ML-based methods for energy consumption prediction

In order to evaluate the effectiveness of the proposed method for energy consumption prediction, five ML-based methods, predominantly ensemble methods, were used, namely RF, DT, XGBoost, LightGBM, and AdaBoost. The details of these methods are described below.

RF: It is an ML-based ensemble method that combines the prediction results of multiple individual decision trees using bagging technique. The execution of the combined process within the method can be explained as follows: First, subsets are created from the training data using random sampling technique. It then trains the decision trees with these subsets. Subsequently, the final prediction result is obtained by averaging predictions from all decision trees (Liaw & Wiener, 2002). This is expressed mathematically using the following steps.

(35) A={(x1,y1),(x2,y2),...,(xn,yn)}

(36) A′={(x′1,y′1),),(x′2,y′2),...,(x′m,y′m)},(m<n).

In Eq. (35), the symbols A, xi, yi denote the training dataset, feature vector, and target output, respectively. Equation (36) describes a random subset comprising n-instances produced by random sampling technique. For each decision tree T, a model is constructed using the dataset A′, with each tree making predictions based on the instance data. Equation (37) shows the class label in a classification context, while Eq. (38) indicates the final prediction value for a regression task.

(37) y~=argmx(∑yl~′==k),k=1,2,3…k

(38) y~=(1/m)∗∑yl∼′.

Here, m is the total number of decision trees, yi~′ is the prediction output of the decision tree, and y~ is the final prediction result of the method.

DT: This method is widely used in many regression tasks because it not only yields highly accurate forecasts within predictive modeling, but also boasts a straightforward and comprehensible framework (Huynh-Cam, Chen & Le, 2021). When predicting data, the basic DT procedure entails the construction of a tree model based on feature values. In this procedure, the data is first divided into subgroups using these feature values. A decision tree is then created for each subgroup of data. The building of decision trees starts at the root node and progresses through the structure, forming branches and leaves according to decisions that hinge on the feature values (Li et al., 2022). During decision making, either the Gini index or entropy metrics are evaluated; with the Gini index being preferable for distinguishing classes that occur more frequently, and entropy being employed to generate a tree structure with better balance. The Gini index serves as an indicator of a feature’s purity in the dataset and can be calculated as shown in Eq. (39).

(39) Giniindex=1−∑in⁡Pi2.

Here Pi2 gives the probability of occurrence for the ith class. The Gini index value ranges between 0 and 1, and its value close to 0 indicates a higher discriminative power of the class. DTs are trained using the Classification and Regression Tree (CART) algorithm in this method.

XGBoost: The algorithm proposed by Chen & Guestrin (2016) represents a sophisticated implementation of the gradient boosting framework, classified as a machine learning-based ensemble method. This method is distinguished by several key attributes, including its ability to mitigate overfitting, handle missing data effectively, deliver high predictive accuracy, and demonstrate rapid processing capabilities (Shehadeh et al., 2021). The basic principle of XGBoost is based on selecting instances and learning from the remnants of the previous model to create a new simple model by minimizing the objective function. Let’s assume X={(xi,yi):xiϵRe,yiϵR|X|=n} is a dataset with n-instances and e-features. In this case, the output prediction for the ensemble tree model can be derived using the joint function K, as shown in Eq. (40).

(40) y^i=θ(xi)=∑k=1K⁡fk(xi).

Here, y^i denotes the predicted value for the ith instances, K defines the number of CART regression trees, fk stands for the prediction of the kth tree for the ith instances, and fkϵF,F={f(x):wq(x),q:Re→T,wϵRT} describes the space where the trees reside. In the equation given for this space, q is the structure of trees, w is the weight of leaves, and T is the number of leaf nodes in trees. For this method, the training process is carried out using the objective function given in Eq. (41), which combines a gradient boosting loss and a regularization term.

(41) L(θ)=∑i⁡l(y^i,yi)+∑k⁡Ωfk.

In the equation, the first part corresponds to the gradient boosting loss, while the second one represents the regularization term (Ren et al., 2023). θ and yi are the parameter of the model and the real value for the ith instances. The symbol l denotes the loss function, measuring the difference between predicted value and real value. The symbol Ω defines the regularization term, indicating the complexity of the model. The value of this term can be computed by the formula given in Eq. (42).

(42) Ω(f)=ςT+1/2λ∥ω2∥.

Here, ς and λ are manually adjusted regulation parameters, T is the number of leaf nodes, and ω is the weight vector associated with all leaf nodes of the decision trees.

LightGBM: This method is characterized as a fast and high-performance ML algorithm that, similar to XGBoost, is based on gradient boosting principles. Its capacity to manage large datasets with minimal resource expenditure, coupled with its swift processing speed, high predictive accuracy, and low memory usage makes it a standout among boosting algorithms (Ke et al., 2017). The functionality of LightGBM is based on a histogram-based approach, which contributes to a shorter training time and efficient resource utilization. Furthermore, this algorithm reduces the computational cost of the method by discretizing continuous-valued data. However, a notable challenge arises when attempting to scan all instances to identify the optimal split, particularly for large datasets. To address the issues of prolonged training times and increased resource demands, the algorithm employs gradient-based one-side sampling (GOSS) and exclusive feature bundling (EFB) techniques (Brownlee, 2021b). GOSS reduces the dataset by scanning only the most critical instances rather than evaluating the entire dataset. Meanwhile, EFB creates dense features by combining sparse features. A key feature that differentiates this method from other tree-based techniques is its leaf-wise learning strategy usage, which focuses on the leaves of trees during training, potentially resulting in improved performance and accelerated learning.

AdaBoost: The method developed by Freund & Schapire (1997) is an ML-based ensemble algorithm widely used for both classification and regression tasks. It combines the outputs of multiple weak predictors through a series of specific steps to create an effective and robust regression model. The underlying idea of this method is based on the iterative training of weak predictors on the same training dataset, followed by the aggregation of their results, which may vary in prediction accuracy (Ren et al., 2022). For this process, initially, AdaBoost adjusts the weights based on the output of each weak predictors by changing the data distribution. Subsequently, the updated weights on this new data are fed as inputs to the next weak prediction model. Ultimately, the final result is derived by combining all the weak predictors. Let us denote the outputs of k weak predictors for the dataset X = { x1, x2, x3,…, xr} are obtained as t1(X),t2(X),…tk(X). In this case, the final output ( T(X)) of the method can be given as follows:

(43) T(X)=sign(∑ik⁡(αiti(X)).

Here, k is the number of trained weak predictors, ti is the prediction result associated with the ith weak predictor, and αi is the weight assigned to the ith weak predictor.

DL-based weighted average ensemble method for energy consumption prediction

In this study, an additional ensemble method based on the DL-based weighted average technique was also employed to verify the performance of the proposed method for predicting energy consumption. This method is structured through an arrangement that entails the removal of the input reducer and prediction combiner units from the proposed method framework illustrated in Fig. 1, substituting them with a combining unit that implements the weighted average technique. The weighted average-based unit obtains the final prediction result by aggregating the scores of DL-based methods from the base predictor unit, as expressed in Eq. (44).

(44) ECtesti=∑j=15⁡BPtestji×wtrnji∑j=15⁡wtrnji.

Here, ECtesti is the energy consumption prediction for the ith fold of data, BPtestji is the score produced by the jth predictor for the ith fold of data, and wtrnji is the weight of the jth predictor for the ith prediction data. The weight values are determined through a two-stage process during the training phase. In the first stage, the average weights are calculated by assessing MAE values as outlined in Eq. (45), based on the scores of the base predictors.

(45) wji(MAEtrn)=1MAEtrnjimax(MAEtrnji)|j=m.

In the given equation above, max(MAEtrnji)|j=m represents the DL-based predictor with the highest MAE value during the training phase for the ith fold of data, while MAEtrnji denotes the MAE value of the jth predictor for the same fold of data. In the second stage, the optimal weight values are derived through experimental studies conducted on the average weight values. Table 3 shows the optimized weights of the outputs from the base predictors, which were obtained as a result of the experimental investigations for the IBSH, HCEC and IHPC datasets.

Table 3 The optimized weights of the outputs from the base predictors obtained as a result of the experimental investigations for the IBSH, HCEC and IHPC datasets.

Dataset	CNN	RNN	LSTM	BI-LSTM	GRU	
w1opt	w2opt	w3opt	w4opt	w5opt	
IBSH	0.0009	0.0724	0.0032	0.5270	0.3964	
HCEC	0.0040	0.0040	0.0120	0.0600	0.9200	
IHPC	0.0005	0.1200	0.6095	0.1300	0.1400	

Experimental results and discussion

This section presents the experimental results and evaluations of the proposed method for predicting energy consumption, along with the ML, DL, and DLBWA ensemble methods used to validate its effectiveness. In addition, it provides comparisons between the performance results of similar studies in the literature and those of the proposed method. For this study, the Python programming language was used due to its rich library support for developing ML and DL-based energy consumption prediction models. Specifically, libraries such as Nymph, Pandas, Matplotlib, Scikit-learn and Kernel libraries, which provide user-friendly features, were utilized. Google Collab was preferred as the development platform for these models because it supported all libraries related to the Python programming language and offered the capability to use accelerators, including graphics processing units (GPUs) and tensor processing units (TPUs). The data preprocessor, five-fold CV data splitter, and base predictor units of the proposed method were created in the Python programming language. Conversely, the ANFIS and input reducer units were implemented in the MATLAB programming language. The choice of MATLAB for ANFIS was driven by the necessity for various types of membership functions for the inputs, which are crucial for determining the optimal structure of this unit within the proposed method. In essence, MATLAB offers us more flexibility and options for working with diverse membership functions and fine-tuning ANFIS. The implementation of the method in two distinct environments was achieved by transferring the base predictor outputs to two separate databases during the training and testing phases. The parameters and their values described in the relevant subheading in “Materials and Methods” were used in the configuration of ANFIS. The ML and DL-based methods used in this study were implemented with the parameters and their setting values specified in Table 4.

Table 4 The parameters used in the ML and DL-based methods and their setting values.

ML based methods	Parameters’ values	DL based methods	Parameters’ values	
RF	n_estimators: 100 criterion: “gini”	GRU	units:100 activation: sigmoid	
max_depth: None min_samples_split: 2	return_sequences: False dropout: 0.0	
min_samples_leaf:1 max_features: “auto”	reset_after: False epoch: 250.	
random_state: 42	optimazer: adam	
DT	criterion: “gini” splitter: “best”	LSTM	units:50 activation: sigmoid	
max_depth: None min_samples_split: 2	return_sequences: False dropout: 0.0	
min_samples_leaf:1 max_features: None	epoch: 250 optimazer: adam	
random_state: None		
XGBoost	n_estimators: 100 learning_rate: 0.3	Bi-LSTM		
max_depth: 6 min_child_weight:1	units: 50 activation: sigmoid	
subsample: 1 colsample_bytree: 1	return_sequences: False dropout: 0.0	
colsample bylevel:1 gamma: 0	epoch: 250 optimazer: adam	
alpha: 0 lambda: 1		
scale_pos_weight: 1early_stopping_rounds30		
LightGBM	boosting_type: “DART” num_leaves: 2**4	CNN	convolution layer: 2 filters: 128	
max_depth:4 learning_rate:0.1	padding: same activation: relu	
n_estimators100 min_child_weight:0.001	kernel_size 1: 5 pool_size 1: 5	
min_child_samples:20 subsample:1.0	kernel_size 2: 4 pool_size 2: 2	
subsample_freq:0 colsample_bytree: 1.0	dropout: 0.2 trainable: True	
reg_alpha: 0.0 reg_lambda: 0.0	batch_size: 128 epoch: 250	
random_state: None	optimazer: adam	
AdaBoost	base_estimator: DecisionTreeClassifier	RNN	units: 50 activation: tanh	
max_depth: 1 n_estimators: 50	return_sequences: False dropout: 0.0	
learning_rate: 1.0	optimazer: adam	

The performance analyses of all methods used for energy consumption prediction were conducted on three datasets using the five-fold cross-validation technique: the IBSH, HCEC, and IHPC datasets.

Table 5 presents the fold performances of the ML, DL, DLBWA ensemble methods, and the proposed method on the IBSH dataset under five-fold cross-validation conditions. The results indicated that the proposed method achieved lower error values than the other methods in terms of the RMSE metric, with scores of 0.000918, 0.000586, 0.001150, and 0.002961 for Folds 1, 2, 3, and 4, respectively. However, in Fold 5, the RF method surpassed the proposed method, yielding a lower RMSE score of 0.001217. Furthermore, when evaluating the results in terms of the MAE metric, the proposed method consistently obtained lower error values compared to the other methods, with scores of 0.000560, 0.000607, and 0.002086 in folds 1, 3, and 4, respectively. For the same metric, the RF method produced similar results for folds 2 and 5, yielding scores of 0.000318 and 0.000349, respectively. The DLBWA ensemble method, which utilizes the same base predictors as the proposed approach, demonstrated lower error values compared to DL-based methods. This method yielded MAE scores of 0.001894, 0.006243, and 0.003159 for Folds 2, 3, and 4, while providing RMSE scores of 0.002496 and 0.007242 for Folds 2 and 3, respectively. Compared with the ML-based methods, it produced lower error values in Folds 3 and 4, with RMSE scores of 0.007242 and 0.009285, respectively, while it provided a lower error value in only Fold 4, with an MAE score of 0.003159. In the remaining folds, the RF method exhibited lower error values than the DLBWA ensemble method in terms of RMSE and MAE metrics.

Table 5 Fold performances of the ML, DL, DLBWA ensemble methods, and the proposed method on the IBSH dataset under five-fold cross-validation conditions.

Method	Fold 1	Fold 2	Fold 3	Fold 4	Fold 5	
RMSE	MAE	RMSE	MAE	RMSE	MAE	RMSE	MAE	RMSE	MAE	
DT	0.003170	0.001165	0.001951	0.000606	0.027601	0.004041	0.078912	0.011703	0.002686	0.000839	
RF	0.002042	0.000635	0.001359	0.000318	0.010363	0.001511	0.071483	0.008529	0.001217	0.000349	
AdaBoost	0.094473	0.071966	0.124520	0.095286	0.129005	0.096302	0.117946	0.080951	0.106489	0.081412	
LightGBM	0.110723	0.098434	0.084465	0.072862	0.132120	0.074247	0.269145	0.119046	0.093849	0.081055	
XGBoost	0.008472	0.005162	0.005040	0.003176	0.013206	0.004649	0.027130	0.007297	0.005674	0.003602	
LSTM	0.031794	0.025147	0.023732	0.017168	0.044978	0.038686	0.131869	0.076412	0.037781	0.033695	
CNN	0.103754	0.080887	0.110504	0.089404	0.184408	0.166694	0.180036	0.109885	0.131577	0.108569	
RNN	0.064940	0.057220	0.033216	0.028414	0.034833	0.022568	0.060661	0.023564	0.033011	0.022257	
GRU	0.025836	0.025743	0.025148	0.025074	0.009941	0.008588	0.010974	0.006434	0.021659	0.021503	
BI-LSTM	0.002734	0.002254	0.015337	0.015298	0.007283	0.007118	0.008028	0.006084	0.031130	0.031098	
DLBWA ensemble	0.007846	0.007343	0.002496	0.001894	0.007242	0.006243	0.009285	0.003159	0.026269	0.026133	
Proposed method	0.000918	0.000560	0.000586	0.000359	0.001150	0.000607	0.002961	0.002086	0.002042	0.001438	

Table 6 illustrates the fold performances of the ML, DL, DLBWA ensemble methods, and the proposed method on the HCEC dataset under five-fold cross-validation conditions. For this dataset, the proposed method attained the lowest error values in terms of RMSE metric, with scores of 0.011284 and 0.005298 in Folds 2 and 4, respectively. In the remaining folds, the lowest error values were achieved by the XGBoost, DT, and DLBWA ensemble methods, with scores of 0.003464, 1.41E−13, and 0.077036. In terms of the MAE metric, the ML-based methods DT and RF achieved scores of 9.29E−14, 0.000676 and 0.000439, 0.003402 respectively, with the lowest error values in two folds. On the other hand, the proposed method secured the lowest error in a single fold, scoring 0.00707. Notably, excluding the ML-based methods, the proposed method outperformed the DL-based and DLBWA ensemble methods regarding these error scores, achieving lower error values in all folds for the MAE metric and in all folds except Fold 5 for the RMSE metric.

Table 6 Fold performances of the ML, DL, DLBWA ensemble methods, and the proposed method on the HCEC dataset under five-fold cross-validation conditions.

Method	Fold 1	Fold 2	Fold 3	Fold 4	Fold 5	
RMSE	MAE	RMSE	MAE	RMSE	MAE	RMSE	MAE	RMSE	MAE	
DT	0.021221	0.000450	0.219003	0.008782	1.41E−13	9.29E−14	0.045017	0.000676	0.379917	0.007431	
RF	0.014452	0.000439	0.042431	0.003402	0.032296	0.000923	0.049519	0.000743	0.379917	0.007431	
AdaBoost	1.462348	1.160943	1.343634	1.106645	1.177600	0.948150	1.164809	0.959333	1.923344	1.565063	
LightGBM	6.087640	5.829399	6.952186	6.702647	5.802473	5.569163	6.759359	6.549570	5.422290	5.204142	
XGBoost	0.003464	0.001217	0.013649	0.003561	0.003735	0.001293	0.046428	0.002146	0.380524	0.008628	
LSTM	0.226415	0.171597	0.113319	0.079707	0.178590	0.155432	0.222924	0.173594	0.207705	0.144414	
CNN	5.105758	4.025421	3.662745	2.842256	4.021031	3.183243	4.729645	4.050129	5.267134	4.254015	
RNN	0.060621	0.043737	0.307155	0.135248	0.519748	0.428924	0.441524	0.374038	0.708068	0.609439	
GRU	0.016781	0.013624	0.026828	0.024624	0.104710	0.104302	0.075747	0.075507	0.095166	0.092375	
BI-LSTM	0.320183	0.319040	0.067497	0.051738	0.184131	0.182141	0.098038	0.094979	0.116056	0.113848	
DLBWA ensemble	0.027172	0.019142	0.026305	0.022039	0.102113	0.100577	0.074497	0.072532	0.077036	0.068176	
Proposed method	0.004806	0.003566	0.011284	0.008048	0.008949	0.006584	0.005298	0.004177	0.095704	0.007070	

Table 7 presents the fold performances of the ML, DL, DLBWA ensemble methods, and the proposed method on the IHPC dataset under five-fold cross-validation conditions. The proposed method achieved the lowest error values in two folds, with RMSE scores of 0.004407 and 0.002788, as well as MAE scores of 0.001833 and 0.002067 for Fold 1 and 3, respectively. In the remaining folds, RF yielded lower error values for both metrics, with RMSE scores of 0.013172, 0.003366, and 0.001557, and MAE scores of 0.002206, 0.000959, and 0.000645 for Folds 2, 4, and 5. Despite the RF method outperforming the proposed method in terms of the number of folds with the lowest error scores, the proposed method demonstrated superior overall performance, achieving average RMSE and MAE scores of 0.013640 and 0.006572, respectively. When comparing the performance of the proposed method to DL-based methods and the DLBWA ensemble method, it was observed that the proposed method produced lower error values across all folds for the MAE metric, and in all folds except Fold 2 for the RMSE metric.

Table 7 Fold performances of the ML, DL, DLBWA ensemble methods, and the proposed method on the IHPC dataset under five-fold cross-validation conditions.

Method	Fold 1	Fold 2	Fold 3	Fold 4	Fold 5	
RMSE	MAE	RMSE	MAE	RMSE	MAE	RMSE	MAE	RMSE	MAE	
DT	0.170138	0.010302	0.021934	0.003604	0.080453	0.017857	0.004858	0.001568	0.002733	0.001052	
RF	0.124032	0.005889	0.013172	0.002206	0.127738	0.036152	0.003366	0.000959	0.001557	0.000645	
AdaBoost	0.711665	0.604313	0.674327	0.566391	1.004549	0.688029	0.746522	0.601687	0.700648	0.580596	
LightGBM	1.527708	0.884048	1.282800	0.917733	1.243580	0.919199	0.988095	0.712693	0.877062	0.674309	
XGBoost	0.046492	0.020559	0.042043	0.023168	0.084190	0.037659	0.030488	0.019992	0.025718	0.017347	
LSTM	0.077363	0.061871	0.043039	0.030708	0.025859	0.016939	0.071589	0.065697	0.030064	0.021032	
CNN	0.640999	0.576902	0.298820	0.239052	0.270965	0.208256	0.591188	0.465042	0.751997	0.614804	
RNN	0.079072	0.046822	0.104518	0.075235	0.081831	0.065026	0.049356	0.043622	0.058636	0.043367	
GRU	0.008343	0.006458	0.031872	0.031430	0.015936	0.013608	0.146196	0.146176	0.021008	0.019997	
BI-LSTM	0.019716	0.014702	0.032322	0.031345	0.067749	0.067103	0.061472	0.061221	0.047928	0.047382	
DLBWA ensemble	0.054848	0.042243	0.033642	0.025183	0.018471	0.011329	0.024577	0.020669	0.017735	0.011565	
Proposed method	0.004407	0.001833	0.035186	0.011556	0.002788	0.002067	0.022432	0.014742	0.003386	0.002664	

Table 8 shows the average fold performances of ML, DL, and DLBWA ensemble methods, along with the proposed method on the IBSH dataset under five-fold cross-validation conditions. The results revealed that the proposed method outperformed the others, achieving lower error scores: an MAE of 0.001010, RMSE of 0.001531, MSE of 0.0000031, and MAPE of 0.001573. When assessing the results for ML methods, XGBoost excelled in RMSE and MSE with lower error scores of 0.011904 and 0.000208, respectively, compared to other ML-based methods, while RF achieved this performance in MAE and MAPE with those of 0.002268 and 0.007074. Among the DL-based methods, BI-LSTM performed the best, with an MAE of 0.012371, RMSE of 0.012902, MSE of 0.000266, and MAPE of 0.027887. The DLBWA ensemble method surpassed the DL-based methods across all metrics and showed superior performance in RMSE and MSE, with lower error scores of 0.010627 and 0.000179, respectively, compared to the ML-based methods. However, in terms of the MAE and MAPE metrics, it lagged behind the ML-based RF method.

Table 8 Average fold performances of the ML, DL, DLBWA ensemble methods, and the proposed method on the IBSH dataset under five-fold cross-validation conditions.

Method	MAE	RMSE	MSE	MAPE	
DT	0.003671	0.022864	0.001402	0.007629	
RF	0.002268	0.017293	0.001045	0.007074	
AdaBoost	0.085183	0.114487	0.013265	0.342915	
LightGBM	0.089129	0.138061	0.023619	0.118587	
XGBoost	0.004777	0.011904	0.000208	0.012886	
LSTM	0.038222	0.054031	0.004483	0.081685	
CNN	0.111088	0.142056	0.021342	0.373647	
RNN	0.030805	0.045332	0.002261	0.064949	
GRU	0.017468	0.018712	0.000398	0.035406	
BI-LSTM	0.012371	0.012902	0.000266	0.027887	
DLBWA ensemble	0.008955	0.010627	0.000179	0.017521	
Proposed method	0.001010	0.001531	0.0000031	0.001573	

Table 9 illustrates the average fold performances of the ML, DL, and DLBWA ensemble methods, along with the proposed method, on the HCEC dataset under five-fold cross-validation conditions. The results reveal that the proposed method demonstrated significantly higher performance in terms of RMSE, MSE, and MAPE, with lower error scores of 0.025208, 0.001884, and 0.000137, respectively, compared to all other methods. Despite outperforming many methods with an MAE error score of 0.005889, it falls short of RF’s performance in this specific metric. When evaluating the results for the ML methods, XGBoost stood out for its superior performance in terms of RMSE and MSE, with lower error scores of 0.089560 and 0.029433, respectively, compared to the other ML methods. Conversely, the RF method excelled in MAE and MAPE, with lower error scores of 0.002588 and 0.000212, respectively. Among the DL-based methods for this dataset, GRU emerged as the top method across all metrics, achieving lower error scores, including an MAE of 0.062086, RMSE of 0.063846, MSE of 0.005352, and MAPE of 0.001035. GRU outperformed the ML-based methods in RMSE and MSE, but lagged behind the RF method in terms of MAE and MAPE. The DLBWA ensemble method exhibited superior performance compared with the DL-based methods, with lower error scores: MAE = 0.056493, RMSE = 0.061425, MSE = 0.004668, and MAPE = 0.000931. However, similar to GRU, it achieved a lower performance than the ML RF method in MAE and MAPE.

Table 9 Average fold performances of the ML, DL, DLBWA ensemble methods, and the proposed method on the HCEC dataset under five-fold cross-validation conditions.

Method	MAE	RMSE	MSE	MAPE	
DT	0.003468	0.133032	0.038955	0.000216	
RF	0.002588	0.103723	0.029968	0.000212	
AdaBoost	1.148027	1.414347	2.077317	0.020131	
LightGBM	5.970984	6.204790	38.830220	0.091380	
XGBoost	0.003369	0.089560	0.029433	0.000226	
LSTM	0.144949	0.189791	0.037767	0.002518	
CNN	3.671013	4.557262	21.153078	0.068418	
RNN	0.318277	0.407423	0.212892	0.005814	
GRU	0.062086	0.063846	0.005352	0.001035	
BI-LSTM	0.152349	0.157181	0.032811	0.002598	
DLBWA ensemble	0.056493	0.061425	0.004668	0.000931	
Proposed method	0.005889	0.025208	0.001884	0.000137	

As seen from the average fold performances listed in Table 10 for the IHPC dataset, the proposed method outperformed the other methods with lower error scores, including MAE = 0.006572, RMSE = 0.013640, MSE = 0.000356, and MAPE = 0.000943. Similarly, when evaluating the results in terms of ML methods, it was observed that the XGBoost method performed better in RMSE and MSE, with error scores of 0.045786 and 0.002522, while the DT method performed better in MAE and MAPE, with error scores of 0.006877 and 0.001999, respectively. For this dataset, among the DL-based methods, LSTM, GRU, and BI-LSTM exhibited better performances in the MAE-MAPE, RMSE, and MSE metrics, respectively. The performances were obtained as an MAE score of 0.039249 and MAPE score of 0.006635 in LSTM, RMSE score of 0.044671 in GRU, and MSE score of 0.002420 in BI-LSTM. Similar to the other datasets, the DLBWA ensemble method exhibited a better performance than the DL-based methods on this dataset across all metrics, with low error scores of MAE = 0.022198, RMSE = 0.029855, MSE = 0.001080, and MAPE = 0.003739.

Table 10 Average fold performances of the ML, DL, DLBWA ensemble methods, and the proposed method on the IHPC dataset under five-fold cross-validation conditions.

Method	MAE	RMSE	MSE	MAPE	
DT	0.006877	0.056023	0.007186	0.001999	
RF	0.009170	0.053973	0.006378	0.004370	
AdaBoost	0.608203	0.767542	0.603701	0.140089	
LightGBM	0.821597	1.183849	1.454305	0.087642	
XGBoost	0.023745	0.045786	0.002522	0.004779	
LSTM	0.039249	0.049583	0.002907	0.006635	
CNN	0.420811	0.510794	0.297720	0.088737	
RNN	0.054815	0.074682	0.005949	0.008767	
GRU	0.043534	0.044671	0.004631	0.007863	
BI-LSTM	0.044351	0.045837	0.002420	0.007924	
DLBWA ensemble	0.022198	0.029855	0.001080	0.003739	
Proposed method	0.006572	0.013640	0.000356	0.000943	

Figure 5 depicts the average energy consumption prediction curves of the proposed method, the DLBWA ensemble method, the two best DL-based methods, and the two best ML-based methods applied to the IBSH dataset for a 120-h (5-day) period. In this context, BI-LSTM and GRU signify the two best DL methods, and RF and XGBoost denote the two best ML-based methods. Analysis in the a–f range given in Fig. 5 reveals that the proposed method achieved the most accurate energy consumption prediction, with an RMSE score of 0.001425. Although the DLBWA ensemble method showed a better energy consumption prediction with an RMSE score of 0.009620 than the DL-based methods, as in the average performance analyses, this result lags behind the performance produced by ML-based methods RF and XGBoost.

Figure 5 Average energy consumption curves of the methods applied to the IBSH dataset for a 120-h period.

(A) GRU, (B) BI-LSTM. (C) Proposed method (D) RF (E) XGBoost (F) DLBWA ensemble method.

Figure 6 shows the average energy consumption prediction curves of the proposed method, DLBWA ensemble method, two best DL-based methods, and two best ML-based methods applied to the HCEC dataset for a 120-h period. From the prediction curves in the a-f range and the RMSE scores for this period, it is clear that the proposed method was closest to predicting the real energy consumption, with an RMSE of 0.003371. Subsequently, ML-based methods RF and XGBoost produced the next best results, with RMSE scores of 0.016958 and 0.008283, respectively. The DLBWA ensemble method obtained lower performance than both ML-based methods RF and XGBoost and the DL-based GRU method, with an RMSE score of 0.077349.

Figure 6 Average energy consumption curves of the methods applied to the HCEC dataset for a 120-h period.

(A) BI-LSTM, (B) GRU, (C) proposed method (D) RF (E) XGBoost (F) DLBWA ensemble method.

Figure 7 shows the average energy consumption prediction curves of the proposed method, DLBWA ensemble method, two best DL-based methods, and two best ML-based methods applied to the IHPC dataset for a 120-h period. Similar to the findings for other datasets, the proposed method achieved the closest prediction to the real energy consumption in this dataset, with an RMSE score of 0.005915. The analysis of both the energy consumption curves and RMSE results reveals that the DLBWA ensemble method, with an RMSE score of 0.025897, reached a prediction closer to the real energy consumption in this dataset than the DL-based methods BI-LSTM and GRU. However, with this result, it falls behind ML-based RF method. Furthermore, the energy consumption curves and RMSE performances confirmed that the two ML-based methods outperformed the DL-based methods, BI-LSTM and GRU.

Figure 7 Average energy consumption curves of the methods applied to the IHPC dataset for a 120-h period.

(A) BI-LSTM, (B) GRU, (C) proposed method (D) RF (E) XGBoost (F) DLBWA ensemble method.

The characteristic structure, quantity, resolution, distribution (split ratio or k-fold cross-validation (CV) conditions), and normalization of the data used in experimental studies play a significant role to accurately assess the performance of the proposed models in time-series applications. In this study, three datasets with different characteristics, resolutions, and quantities were used to evaluate the performance of the proposed DLBNE method with BST-ANFIS. In addition, a reliable training process for the method was implemented through data normalization. Also, the effectiveness of the proposed method was validated against the ML and DL approaches. In the literature, the IBSH and IHPC datasets included in this study were used together with FTS, ML, and DL-based energy consumption prediction approaches. Current studies have mainly focused on DL and hybrid DL-based energy consumption approaches using these datasets. Table 11 presents a detailed analysis of the RMSE performance comparisons based on the data normalization, resolution, and split ratios of the methods used in previous studies that include these two datasets with the proposed method.

Table 11 Comparison of the proposed method with the methods used in studies involving the same datasets in our study.

Author	Dataset	Methodology	Normalization/Resolution	Data splitting conditions	Performance results (RMSE)	
Vinicius Bitencourt et al. (2021)	IHPC	PCA -WMVFTS	No/IHPC: 1 and 30-min	0.75	IHPC (1-m): PCA-WMVFTS: 0.067	
IBSH (1-m): PCA-WMVFTS: 0.024	
IBSH	IBSH: 1 and 10-min	IHPC (30-m): PCA-WMVFTS: 0.069	
IBSH (10-m): PCA-WMVFTS: 0.04	
Iram et al. (2023)	IBSH	SVM, LR, RF, DT, KNN, NN	No/1-min	0.9943 (Traning: 348-day Test: 2-day)	SVM: 0.19 DT: 0.39	
LR: 0.53 KNN: 0.21	
RF: 0.26 NN: 0.14	
Abdel-Basset et al. (2022)	IHPC	STLF-Net	No/1-H	0.75	0.4386	
Saad Saoud, AlMarzouqi & Hussein (2022)	IHPC	Multistep transformer-SWT	No/1-min, 1-H, 1-Day, 1-Week	–	1-min 1-H	
0.3929 0.4183	
1-Day 1-Week	
0.2378 0.1574	
Gonçalves, Ribeiro & Pereira (2023)	IHPC	ConvLSTM2D VSCA	Yes/1-Day	0.8	ConvLSTM2D: 0.12434	
VSCA: 0.11831	
AL-Ghamdi, AL-Ghamdi & Ragab (2023)	IHPC	DNN multilayered LSTM	Yes/1-min	0.7	0.02410	
Mubarak et al. (2024)	IHPC	LSTM-Attention	Yes/1-H, 1-Day	0.8	1-H 1-Day	
0.0256 0.01801	
Mocanu et al. (2016)	IHPC	FCRBM	No/15-min
1-H, 1-Day	–	15-min 1-H 1-Day	
0.6211 0.6663 0.8286	
Bhoj & Singh Bhadoria (2022)	IBSH		No/1-H, 1-Day	0.75	1-H 1-Day	
CNN-LSTM	CNN-LSTM: 0.346 0.228	
CNN-GRU	CNN-GRU: 0.344 0.202	
Priyadarshini et al. (2022a)	IBSH	ARIMA, SARIMA, LSTM, Prophet, Light GBM, VAR	No/1-min	0.8	ARIMA: 0.1806 Prophet: 0.2666	
SARIMA: 0.2002 LightGBM: 0.2020	
LSTM: 0.2808 VAR: 0.2684	
Kim & Cho (2019b)	IHPC	CNN-LSTM	No/1-min, 1-h, 1-Day, 1-Week	10-fold CV	1-min 1-H	
0.6114 0.5957	
1-Day 1-Week	
0.3221 0.3085	
Proposed method	HCEC	DLBE method with BST-ANFIS	Yes/1-H	5-fold CV	IBSH: 0.001469	
IHPC	IHPC: 0.012442	
IBSH		

When evaluating the results of the studies in Table 11 from the perspective of the data normalization process, it is generally observed that the studies where this process was applied demonstrated a higher performance in terms of the RMSE metric than those where it was not. The normalization process was only applied to the IHPC dataset in studies other than the proposed method, and these are hybrid DL-based approaches. The best performance was achieved with the DLBNE method with BST-ANFIS on the normalized IHPC dataset with a 1-h resolution under five-fold cross-validation condition, resulting in an RMSE score of 0.012442. Conversely, the lowest performance was obtained from experimental studies conducted on the IHPC dataset with 1-day data resolution under 0.8 data splitting conditions using the two-dimensional convolutional long short-term memory (ConvLSTM2D) with Attention (VSCA) method proposed by Gonçalves, Ribeiro & Pereira (2023), resulting in an error score of 0.11831. In a recent study similar to this study, Mubarak et al. (2024) applied a hybrid method that included LSTM and attention mechanisms to normalized IHPC datasets with 1-h and 1-day resolutions, testing the model’s effectiveness in energy consumption prediction with experiments conducted under splitting conditions of 0.8. They demonstrated that their method achieved the best performance after our proposed method, with an RMSE score of 0.01801 from the 1-day resolution IHPC dataset. They also validated that their model could be successfully used for electricity consumption prediction, achieving an RMSE score of 0.0256 from the 1-h resolution dataset. The DNN multilayered LSTM method proposed by AL-Ghamdi, AL-Ghamdi & Ragab (2023) is also a model that can compete with the LSTM-attention method in terms of performance. They applied the proposed model to a normalized IHPC dataset with a 1-min resolution under 0.7 data splitting condition and obtained an RMSE score of 0.02410.

In studies where the data normalization process was not applied, a series of approaches involving FTS, ML, DL, and hybrid methods were used on the IBSH and IHPC datasets with different resolutions. Vinicius Bitencourt et al. (2021) achieved successful results using the PCA-WMVFTS method on IBSH and IHPC datasets. In their experiments conducted under a 0.75 data splitting condition, they obtained RMSE scores of 0.024 and 0.04 for the 1 and 10-min resolution IBSH datasets, respectively, and scores of 0.067 and 0.069 for the 1 and 30-min resolution IHPC datasets, respectively. These scores represent the best results obtained methodologically for the unnormalized dataset class. Iram et al. (2023) and Priyadarshini et al. (2022a) applied ML and statistics-based methods on the 1-min resolution IBSH dataset, achieving RMSE scores ranging from 0.14 to 0.53. Among these methods, the NN method demonstrated the best performance, with an error score of 0.14. On the other hand, there are experimental studies that have implemented some hybrid DL methods, including CNN-LSTM and CNN-GRU, on the unnormalized IBSH and IHPC datasets. Bhoj & Singh Bhadoria (2022) applied the CNN-LSTM and CNN-GRU hybrid DL methods to the 1-h and 1-day resolution IBSH datasets, obtaining RMSE values ranging from 0.202 to 0.346. In experiments conducted under the 0.75-data splitting condition, the best score was achieved using the CNN-GRU method on the 1-day dataset. In a similar study by Kim & Cho (2019b), an experimental study was conducted by applying the CNN-LSTM method to IHPC datasets with four different resolutions under 10-fold cross-validation conditions, resulting in an RMSE score ranging from 0.3085 to 0.6114. They achieved the best score using a dataset with the 1-week resolution. Abdel-Basset et al. (2022) proposed a two-stream deep network for short-term load forecasting (STLF-Net), testing its performance on the 1-h resolution IHPC dataset and obtaining an RMSE performance of 0.4386. Saad Saoud, AlMarzouqi & Hussein (2022) developed a hybrid model based on a stationary wavelet transform (multistep transformer-SWT), which differs from commonly used hybrid prediction models like CNN-LSTM in time series problems. They demonstrated that their method, applied to IHPC datasets with different resolutions, achieved a performance superior to other hybrid DL methods, including CNN-LSTM and CNN-GRU, with an RMSE score of 0.1574 from the 1-week resolution dataset. The energy-consumption prediction model proposed by Mocanu et al. (2016), FCRBM, was applied to IHPC datasets with 15-min, 1-h, and 1-day resolutions, producing error scores of 0.6211, 0.6663, and 0.8286, respectively, making it the lowest performing model in terms of the RMSE metric among all models.

As a result, the DLBNE method with the BST-ANFIS in this study demonstrates superior performance to those in similar studies that utilize these two datasets, achieving RMSE scores of 0.001469 and 0.012442 for the IBSH and IHPC datasets, respectively. From the performance of the methods using the normalization process in Table 11, it is clear that this process contributes significantly to improving the performance of energy consumption prediction models. Moreover, the comparison results show that the RMSE score decreases as the data resolution increases from minutes to weeks, particularly for DL-based methods.

In particular, the results from the average fold and periodic performance analyses show that the proposed BST-ANFIS-based DLBNE method has lower error values than the methods used in this study and those reported in the literature. The low error values of the performance metrics in the proposed method suggest that the predicted energy consumption values are close to the actual energy consumption values, indicating higher accuracy. Therefore, the proposed method is a high-accuracy predictor that can make significant contributions to energy management systems and existing applications in its domain. These contributions include: first, increasing the accuracy and reliability of energy consumption predictions, which enables more effective implementation of energy management strategies; second, optimizing energy consumption planning and reducing energy waste, leading to more efficient use of energy resources and prevention of unnecessary energy consumption; third, facilitating accurate calculation of energy costs, contributing to reduced energy bills and cost savings; fourth, ensuring stable and reliable operation of the energy management system, enabling better response to sudden energy demands and prevention of power outages; and finally, enabling better prediction of maintenance and repair needs for system components, which reduces maintenance costs and extends the system’s lifespan.

Conclusion

In this study, the DLBNE method with BST-ANFIS, which combines the outputs of multiple DL models with the ANFIS architecture through an approach based on the best score transfer, is proposed as an effective model for energy consumption prediction. The novelty of the study lies in its methodological basis, which involves the integration of multiple model outputs into ANFIS through the best score transfer approach. This approach makes four key contributions to the DLBNE method. First, it allows for a structurally simpler configuration of the ANFIS architecture by reducing the number of inputs from the base predictors to the predictor combiner based on the MAE and AE metrics. Second, it improves the performance of the method by effectively combining the two outputs with the best scores among the outputs of base predictors using the ANFIS architecture. Thirdly, it minimizes the reflection of problems arising from the overfitting of base predictors in stacking ensemble models to the output of the meta-combiner. Finally, it makes the method more dynamic by directly obtaining the two best-scoring inputs based on the real-time data flow between multiple models and ANFIS.

Experimental studies were carried out under five-fold cross-validation conditions by applying DLBNE with BST-ANFIS, DL, ML, and DLBWA ensemble methods to the publicly available IBSH, IHPC, and HCEC datasets through three-stage analyses: fold, average, and periodic performance. In the fold analyses on the IBSH dataset, the proposed method performed better than the other methods in terms of the RMSE metric in four-folds and the MAE metric in three folds, while RF showed higher performance in the remaining folds for these metrics. Although the proposed method achieved higher performance than the other methods on the HCEC dataset in terms of the RMSE metric in two-fold and MAE metric in one-fold, its performance was observed to be lower than that on the IBSH dataset when evaluated on a per-fold basis. On the other hand, for this dataset, each of the XGBoost, DT, and DLBWA ensemble methods exhibited higher performance than the proposed method and other methods in terms of the RMSE metric in one-fold, while the RF and DT methods performed better in terms of the MAE metric in two-fold. For the IHPC dataset, the performance of the proposed method was similar to that of the HCEC dataset. For this dataset, when comparing the methods with each other in terms of both metrics, the proposed method achieved higher performance in two folds, while RF outperformed it in three folds. The average performance analyses revealed that the proposed method surpassed the other methods in terms of MAE, RMSE, MSE, and MAPE metrics for almost all datasets. Notably, this remarkable achievement was observed in all metrics except MAE in the HCEC dataset, and in all metrics in the IBSH and IHPC datasets. Although the proposed method demonstrated good performance in a few folds in terms of RMSE and MAE metrics on the fold performance analyses, particularly for the IHPC and HCEC datasets, the average performance analysis results indicated that it could exhibit robust performance in predicting energy consumption across all datasets. Consistent with the other performance results, the proposed method demonstrated the highest performance among all the methods for energy consumption prediction in the 120-h periodic analyses. Similarly, the literature review also indicated that the proposed method had a high performance. Consequently, its effective performance on datasets representing diverse energy consumption conditions may make it a critical component in managing energy for smart homes and individual households.

Although the statistical performance effectiveness of the proposed method has been proven for three different datasets, it still comes with some limitations that need to be addressed in future studies. First, the fact that the proposed method operates on two separate databases due to using two different programming platforms to determine the most appropriate ANFIS configuration reduces its operating speed, limiting its usability in real-time applications. As this study experimentally determined the most appropriate ANFIS configuration, its usability in real-time applications can be enabled by reconstructing the entire model on a single programming platform with these optimized configuration parameters.

Electric energy consumption data exhibit a characteristic structure that combines many factors, such as social, economic, and environmental impacts. Therefore, another limitation of this method is that it relies heavily on specific regional data, affecting its generalizability. However, this issue can be addressed by incorporating a wider range of energy consumption data from various regions to make the method more robust. Moreover, hyperparameter optimization of the base DL predictors can also increase the model robustness and generalizability of electrical energy consumption predictions.

The method includes five different DL approaches as its base learners as well as two training phases, making it structurally complex and computationally expensive in terms of energy consumption prediction. These constraints complicate the applicability of this method, which has a strong performance in terms of error metrics in real-time applications, similar to its first limitation. The following two improvements to the proposed method can devastate the aforementioned limitations through a comprehensive evaluation of the results obtained from ablation studies conducted on base DL learners with increased data diversity. Based on their performances, the first improvement in the ablation studies involves reducing the number of DL approaches used as the base learners from five to three. The second one entails directly applying the prediction result of the DL approach with the best MAE performance obtained from these studies as the first input to ANFIS. Consequently, the task of the unit performing the best score approach in the method is reduced, and eliminating the first training phase used in obtaining the first ANFIS input ensures that the entire model is trained in a single phase. Through both improvements proposed, it is possible to achieve a less complex model with lower computational costs and faster execution compared to its previous form, facilitating high-performance applications capable of real-time operation in residential energy consumption.

Supplemental Information

Supplemental Information 1 MATLAB files for an Adaptive Neuro-Fuzzy Inference System (ANFIS) architecture trained under 5-fold cross-validation conditions based on the DL-based base predictor scores generated during the training phase-I using the “Homedata.ipynb” Jupyter Notebook f.

Supplemental Information 2 Python code that includes a series of preprocessing steps performed before applying DL and ML-based methods mentioned in the study.

The datasets obtained by running the code in each document are as follows: Homedata.csv, Homestead.csv ve Household.csv.

Supplemental Information 3 A MATLAB m-file function that calculates the Mean Absolute Error (MAE).

Supplemental Information 4 Jupyter Notebook files that encompass the entire content of a Notebook created with a Jupyter Notebook web application session.

Each file contains Python code files created for the implementation of deep learning methods like LSTM, CNN, RNN, GRU, and Bi-LSTM for their respective datasets. Each file contains training prediction results, test prediction results, and the results of performance metrics obtained from the application of deep learning methods.

Supplemental Information 5 Python code files created for the implementation of machine learning methods including XGBoost, Decision Trees (DT), LightGBM, Random Forest (RF), and AdaBoost for the “Homedata,” “Homestead,” and “Household” datasets.

Each of these files contains training prediction results, test prediction results, and the results of performance metrics obtained from the application of these machine learning methods.

Supplemental Information 6 A MATLAB M-file program that performs the combining process based on the weighted average of scores in the DLBWA (Deep Learning-Based Weighted Average) ensemble method using the test data from the “Real-time test data scores” database.

This also calculates the fold-based performances.

Supplemental Information 7 A MATLAB M-file program that combines the input decision-making procedure of the proposed method with the ANFIS (Adaptive Neuro-Fuzzy Inference System) architecture.

This obtains the fold-based final results based on the applied test data and calculates the performances.

Supplemental Information 8 Three datasets that have been preprocessed.

Supplemental Information 9 Real-time test data scores database on the proposed schema.

Test files in the “.xlsx” format, containing the scores of DL-based basic predictors produced during the testing phase using the “Homedata.ipynb” Jupyter Notebook file. These files are later used in the Testing phase process.

Supplemental Information 10 Training data scores database on the proposed schema.

The scores of DL-based basic predictors produced during the training phase-I using the “Homedata.ipynb” Jupyter Notebook file in the “dl_app.rar” folder. These files, created under 5-fold cross-validation conditions, are subsequently used in the Training Phase II and Test Phase processes.

Supplemental Information 11 IHPC dataset.

Supplemental Information 12 HCEC dataset.

Supplemental Information 13 IBSH dataset.

Additional Information and Declarations

Competing Interests

The authors declare that they have no competing interests.

Author Contributions

Birce Dağkurs conceived and designed the experiments, performed the experiments, analyzed the data, performed the computation work, prepared figures and/or tables, and approved the final draft.

İsmail Atacak conceived and designed the experiments, performed the experiments, analyzed the data, performed the computation work, prepared figures and/or tables, authored or reviewed drafts of the article, and approved the final draft.

Data Availability

The following information was supplied regarding data availability:

The IBSH dataset is available at Kaggle: https://www.kaggle.com/datasets/taranvee/smart-home-dataset-with-weather-information/code.

The IHPC dataset is available at the UCI Machine Learning Repository: https://archive.ics.uci.edu/dataset/235/individual+household+electric+power+consumption.

The HCEC dataset is available at Kaggle: https://www.kaggle.com/datasets/unajtheb/homesteadus-electricity-consumption.

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
