# Peer review of "Deep learning-based novel ensemble method with best score transferred-adaptive neuro fuzzy inference system for energy consumption prediction"

_PeerJ Computer Science, doi:10.7717/peerj-cs.2680_

## Round 0.1 · original submission · Major Revisions

Dear authors,

Thank you for your interest in submitting your research to the journal PeerJ Computer Science. After the collected review reports for your paper, we will give you the opportunity to revise the paper. Please provide the cover letter with clear reply to each reviewer.

**Language Note:** PeerJ staff have identified that the English language needs to be improved. When you prepare your next revision, please either (i) have a colleague who is proficient in English and familiar with the subject matter review your manuscript, or (ii) contact a professional editing service to review your manuscript. PeerJ can provide language editing services - you can contact us at [email protected] for pricing (be sure to provide your manuscript number and title). – PeerJ Staff

·

Basic reporting

The paper discusses a novel deep learning-based energy consumption prediction method, referred to as the DLBE method, which combines DL-based methods and an ANFIS for energy consumption prediction. The study uses several publicly available datasets (IBSH, IHPC, and HCEC) for experimental analysis and performance evaluation. The paper is well-structured and provides a detailed analysis of the proposed DLBE method's performance, including comparisons with other machine learning methods. However, there are some issues related to clarity and completeness that need attention:
• The paper lacks some crucial details, such as the technical specifics of the DLBE method and ANFIS architecture. Readers may benefit from a more in-depth description of these components.
• The text mentions “the best score inputs of ANFIS” but the criteria for selecting these inputs are not clearly defined.
• The use of terminology and acronyms without prior explanation might make it difficult for non-experts to follow the content.

Experimental design

The experimental design of the study is sound, and the authors provide comprehensive details about the datasets, the cross-validation methodology, and the metrics used for evaluation. The study successfully demonstrates the proposed method's performance across three datasets. However, there are a few issues that should be addressed:
• While the text mentions a 5-fold cross-validation method, the specific details of how data is divided and the parameters used in this process are missing. Providing more transparency about this process would enhance the study's reproducibility.
• A more detailed explanation of the DL-based methods used in the comparisons, such as CNN, RNN, LSTM, BI-LSTM, and GRU, would be helpful for readers unfamiliar with these models.
• It is recommended that the authors provide more insights into the motivation for choosing these specific datasets, as well as the potential real-world applications of the DLBE method.

Validity of the findings

The findings presented in the paper appear to be valid based on the experiments conducted. The authors use a variety of performance metrics, such as RMSE and MAE, to evaluate the proposed DLBE method in comparison to other machine learning methods. These findings show that the DLBE method performs competitively or outperforms other methods in most cases. However, there are some points to consider:
• The paper discusses the results and performance metrics but does not provide substantial insights into the practical implications of these findings for energy management systems or other applications.
• The significance of the improvements or differences in performance, such as “significantly higher performance” could be better supported with statistical analysis or more context.

Additional comments

The paper presents an interesting approach to energy consumption prediction using a combination of DL and ANFIS. It has the potential to contribute to the field of energy management and prediction. However, several improvements are needed for clarity, transparency, and completeness.
• Adding more technical details and algorithmic descriptions of the DLBE method and ANFIS architecture would benefit readers.
• Clarifying the criteria and rationale for selecting inputs in ANFIS would improve understanding.
• Providing more context on the practical implications and applications of the findings would be valuable.
• Addressing issues with terminology, acronyms, and dataset selection motivation will enhance the paper's accessibility and relevance.

Reviewer 2 ·

Basic reporting

a) Motivation and model setup: Proper, convincing, and timely motivation cases should be included.
b) There are no methodological innovations. This paper does not fulfill the minimal PeerJ Comp Sci requirements in terms of originality.
c) The literature review is weak. Gaps were not defined well.
d) It was not elaborated on how the presented research was placed on the existing literature.

Experimental design

a) It was not elaborated on the similarities and differences with the previous research work(s) that were(were) closest to the current one.
b) Research implications were not outlined well. Also, an insight into how this study might impact current trends of research in the area is missing.

Validity of the findings

It is crucial to check whether the findings can challenge/support the current industrial practice (i.e., energy consumption prediction).

Additional comments

1) The writing is not appropriate for the PeerJ Comp Sci community. There are many inflated phrases and unproven statements.
2) The presentation of the work needs thorough revision. Much irrelevant material, mostly lengthy tables, needs removal from the manuscript.

Reviewer 3 ·

Basic reporting

Thank you for inviting me as a reviewer for manuscript titled Deep learning-based novel ensemble method with best score transferred-adaptive neuro fuzzy inference system for energy consumption prediction. The paper presents application of adaptive neuro fuzzy logic system for energy consumption prediction. The paper is impressive for the efforts made from you to demonstrate the valence of your algorithm. The model is well explained. Methodology is clear.
Specific comments
- Need to better highlight the novelty of study in the introduction. Introduction should be clearly stated research questions and targets first. Then answer several questions:
Why is the topic important (or why do you study on it)?
What are the research questions?
What are your contributions?
- The applicability of the method. Why do we need application FLS type 1 in this study? Why nor FLS type 2? I did not see the author discussing the reason. Therefore, it is impossible to prove the superiority of this model combination in this article. Need detailed further explanation.
- Why you use Gaussian type MFs? Please explain benefits of using Gaussian MFs over other types of MFs.
- Conclusion should be extended by highlighting the study novelty. More future directions should be presented. Show limitations of the proposed algorithms.
I will review the final version of the paper with pleasure.

Experimental design

See the above comments!

Validity of the findings

See the above comments!

Additional comments

See the above comments!

---

## Round 0.2 · Minor Revisions

Dear authors,

Your revised paper has been reviewed and we need additional revisions explained in the comments by the reviewer. Please improve the paper and provide a cover letter with replies to the reviewer point to point.

·

Basic reporting

Fine

Experimental design

Good

Validity of the findings

Very good

Additional comments

The authors revised the paper as per comments, accepted

Reviewer 3 ·

Basic reporting

no comment

Experimental design

no comment

Validity of the findings

no comment

Additional comments

The paper looks much better now, however, I still have some minor comments that should be addressed before a positive response:
- Comparison can be made effective with more results and discussion.
- Addressing your research limitations could enhance the credibility, applicability, and impact of your research. It is important to note that limitations in a research paper do not necessarily imply negative aspects but rather areas that offer opportunities for further refinement and improvement. Identifying and discussing these limitations transparently can contribute to the overall growth and effectiveness of the study. Explicitly acknowledge the limitations of the proposed framework and model. Address any potential drawbacks or constraints and how they were managed or could be improved in future iterations.

---

## Round 0.3 · accepted · Accept

Dear authors,

Your paper has been reviewed again, and all reviewers accepted your paper.

Reviewer 3 ·

Basic reporting

All the reviewers' comments have been addressed carefully and sufficiently, the revisions are rational from my point of view, I think the current version of the paper can be accepted.

Experimental design

All the reviewers' comments have been addressed carefully and sufficiently, the revisions are rational from my point of view, I think the current version of the paper can be accepted.

Validity of the findings

All the reviewers' comments have been addressed carefully and sufficiently, the revisions are rational from my point of view, I think the current version of the paper can be accepted.